# Unravelling genomic drivers of speciation in *Musa* through genome assemblies of wild banana ancestors

Guillaume Martin [1,2,11] ✉, Benjamin Istace [3,11], Franc-Christophe Baurens [1,2], Caroline Belser [3], Catherine Hervouet [1,2], Karine Labadie [4], Corinne Cruaud[4], Benjamin Noel [3], Chantal Guiougou[2,5], Frederic Salmon[2,5], Joël Mahadeo[2,6], Fajarudin Ahmad[7], Hugo A. Volkaert [8,9], Gaëtan Droc [1,2], Mathieu Rouard [2,10], Julie Sardos[2,10], Patrick Wincker [3], Nabila Yahiaoui[1,2], Jean-Marc Aury [3,12] & Angélique D'Hont [1,2,12] ✉

Hybridization between wild *Musa* species and subspecies from Southeast Asia is at the origin of cultivated bananas. The genomes of these cultivars are complex mosaics involving nine genetic groups, including two previously unknown contributors. This study provides continuous genome assemblies for six wild genetic groups, one of which represents one of the unknown ancestor, identified as *M. acuminata* ssp. *halabanensis*. The second unknown ancestor partially present in a seventh assembly appears related to *M. a.* ssp. *zebrina*. These assemblies provide key resources for banana genetics and for improving cultivar assemblies, including that of the emblematic triploid Cavendish. Comparative and phylogenetic analyses reveal an ongoing speciation process within *Musa*, characterised by large chromosome rearrangements and centromere differentiation through the integration of different types of repeated sequences, including rDNA tandem repeats. This speciation process may have been favoured by reproductive isolation related to the particular context of climate and land connectivity fluctuations in the Southeast Asian region.

The tropical South Asia region stands out as one of the world's most species-rich areas, a biodiversity hotspot shaped by its intricate tectonic and climatic history[1]. Bananas are native from this region and belong to the genus *Musa* which is estimated to comprise 70 species[2]. Natural inter(sub)specific hybridization between different wild *Musa* species and subspecies[3–6] is at the origin of current banana cultivars. These cultivars are vegetatively propagated and can be diploid, triploid and sometimes tetraploid. Banana domestication occurred several thousand years ago with parthenocarpy and sterility as main targets[3,7] which ensured the production of seedless edible fruits.

[1]CIRAD, UMR AGAP Institut, Montpellier, France. [2]UMR AGAP Institut, Univ Montpellier, CIRAD, INRAE, Institut Agro, Montpellier, France. [3]Génomique Métabolique, Genoscope, Institut François Jacob, CEA, CNRS, Univ Evry, Université Paris-Saclay, Evry, France. [4]Genoscope, Institut François Jacob, CEA, CNRS, Univ Evry, Université Paris-Saclay, Evry, France. [5]CIRAD, UMR AGAP Institut, Capesterre-Belle-Eau, France. [6]CIRAD, UMR AGAP Institut, CRB-PT, Roujol Petit-Bourg, France. [7]Research Center for Applied Botany, Organization Research for Live Sciences and Environment, BRIN, Bogor, Indonesia. [8]Center for Agricultural Biotechnology, Kasetsart University Kamphaengsaen Campus, Nakhon Pathom, Thailand. [9]Center of Excellence on Agricultural Biotechnology (AG-BIO/MHESI), Bangkok, Thailand. [10]Bioversity International, Parc Scientifique Agropolis II, Montpellier, France. [11]These authors contributed equally: Guillaume Martin, Benjamin Istace. [12]These authors jointly supervised this work: Jean-Marc Aury, Angélique D'Hont. ✉e-mail: guillaume.martin@cirad.fr; angelique.dhont@cirad.fr

Nowadays, bananas are popular fruits worldwide and a vital staple food in many tropical and subtropical countries.

Recent advances in banana genomics have led to considerable progress in our understanding of the complex origin of banana cultivars. They revealed that cultivar genomes are complex mosaics involving several species and subspecies. Based on these ancestral genome mosaic and on cultivars geographical distribution, the initial steps of banana domestication were proposed to involve hybridizations between *M. acuminata* ssp. *banksii* (and possibly ssp. *zebrina*) and *M. schizocarpa* in New Guinea[8]. Then, during the diffusion of early cultivars throughout Southeast Asia, additional hybridizations occurred with other local wild *Musa* species and subspecies, including *M. a.* ssp. *zebrina* (in Java), *M. a.* ssp. *malaccensis* (Malayan Peninsula, Sumatra), *M. a.* ssp. *burmannica* (in Southern Indo-Burma), *M. balbisiana* (in India to South China and probably up to the Philippines) and two unknown contributors. These successive steps of hybridization resulted in cultivars with increasing genome complexity obtained from recombination between three to up to seven of nine ancestral contributors, including 4 subspecies of *M. acuminata* (A genome, $2n = 2x = 22$), *M. balbisiana* (B genome, $2n = 2x = 22$) and *M. schizocarpa* (S genome, $2n = 2x = 22$) and two unknown ancestors[8–11] (Fig. 1). In addition, a peculiar small group of Fe'i cultivars from Oceania derived only from *Musa* species of the former Australimusa section (T genome, $2n = 2x = 20$). Large chromosomal rearrangements, i.e. reciprocal translocation and/or inversion, were described in some of these contributing species or subspecies and were transmitted to many cultivars[12–15].

This context of inter(sub)specific hybridization between (sub)species bearing chromosomal rearrangements may have favoured the production of 2x gametes leading to the formation of triploid cultivars[3,4]. The main cultivars include diploids and triploids with various global genomic constitutions (e.g. AA, AB, AAA, AAB, ABB, AAT) modulated by interspecific recombination and introgression from *M. schizocarpa*[5,8,13,16,17]. Only a very limited number of these cultivars are grown on a large scale, with the most highly grown being the AAA Cavendish dessert bananas, the AAA East African Highland cooking bananas and the AAB plantain bananas, respectively representing around 57%, 10% and 16% of the world production[18]. Each of these cultivar groups includes somaclonal phenotypic variants derived from one original seed and centuries or millennia of vegetative propagation.

Since the first *M. a.* ssp. *malaccensis* genome assembly[19], and the further refined telomere-to-telomere version[20], other *Musa* genome assemblies have been reported. Sequencing efforts first focused on wild species and included preliminary assemblies of *M. a.* ssp. *zebrina*, *M. a.* ssp. *burmannica* and *M. a.* ssp. *banksii*[21], a reference genome assembly of *M. balbisiana*[22], a high continuity genome assembly of *M. schizocarpa*[23] and a haplotype resolved assembly of a diploid *M. a.* ssp. *malaccensis* accession[24]. Other assemblies of more distant wild relatives were published, including *M. itinerans*[25], *Ensete glaucum*[26], *M. beccarii*[27] and *M. textilis* (Abaca)[28]. Regarding cultivated bananas, very recently, haplotype-resolved assemblies of a few triploid cultivars including Cavendish have been reported[29–31].

Here, we produce chromosome-scale assemblies for five wild *Musa* accessions and a haplotype resolved one for a cultivar involving the two unknown ancestors. These assemblies complement previously published data and grant access to the genomes of all main ancestral contributors to major banana cultivars. They are used to perform phylogenetic analyses to gain insight into the origin of the unknown ancestors. Using these chromosome-scale assemblies including their complex centromeric sequences, genome-wide analyses are conducted to explore chromosome evolution and speciation mechanisms within this diverse sub-species complex. Finally, these assemblies are used to evaluate recent assemblies of the triploid Cavendish cultivar and to show how they could guide improvement of these assemblies.

## Results
### Overview of chromosome-scale genome assemblies
We assembled the genomes of four wild accessions representing three *M. acuminata* subspecies (ssp. *zebrina*, ssp. *burmannica* and ssp. *banksii*) and *M. textilis*. Moreover, we improved an existing assembly of *M. schizocarpa*. These assemblies were sequenced with a combination of short and long reads, respectively using Illumina and Oxford Nanopore (ONT) technologies. The long-read assemblies were polished using both short and long reads, supplemented by the integration of long-range data. Optical maps and/or Hi-C methodologies were used leading to the production of chromosome-scale assemblies for each genome. We sequenced the genome of Pisang Madu, a diploid hybrid cultivar, using HiFi reads from Pacific Biosciences technology, with the aim of attaining independent assemblies for both haplotypes.

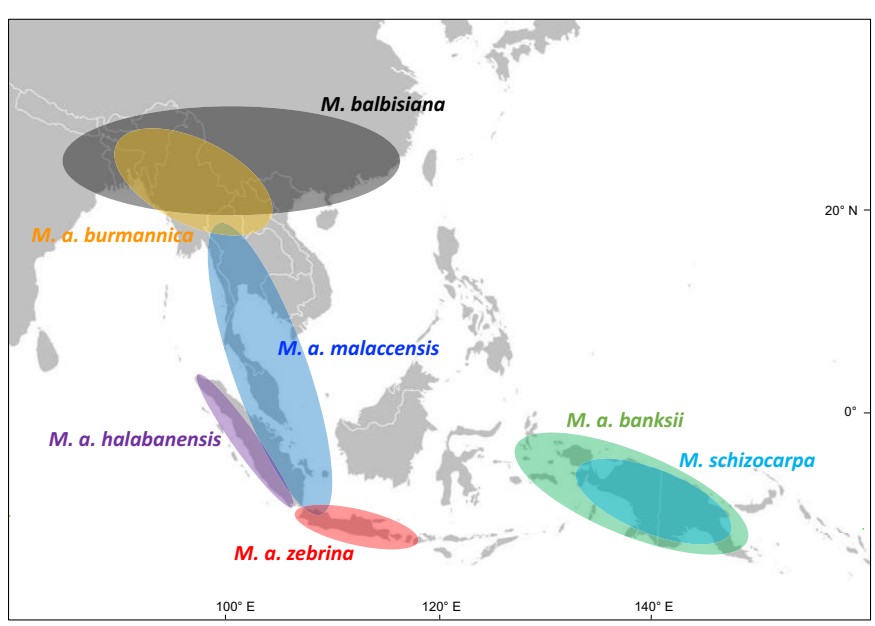

**Fig. 1 | Distribution of the main wild genetic groups that contributed to banana cultivars.** Adapted from De Langhe et al.[90] and Ahmad[33].

**Table 1 | Statistics of the genome assemblies and repeat content**

| | Wild *M. acuminata* ssp. | | | *M. schizocarpa* | *M. textilis* | *Musa* cultivar | |
|---|---|---|---|---|---|---|---|
| | *M. a. zebrina* | *M. a. banksii* | *M. a. burmannica* | | | Pisang Madu H1 (*M. a. halabanensis*) | Pisang Madu H2 |
| Basic chromosome number | 11 | 11 | 11 | 11 | 10 | 11 | 11 |
| Number of sequences* | 14 | 14 | 14 | 14 | 13 | 14 | 14 |
| N50 (Mb) scaffolds (L50) | 49.3 (6) | 45.1 (6) | 44.2 (6) | 46.3 (6) | 52.1 (5) | 45.0 (6) | 47.0 (6) |
| N50 (Mb) contigs (L50) | 8.3 (21) | 1.5 (64) | 3.7 (36) | 20.7 (11) | 4.0 (35) | 18.1 (10) | 10.7 (18) |
| Cumulative assembly size (Mb) | 550.9 | 484.8 | 505.1 | 534.2 | 545.6596533 | 529.6 | 544.6 |
| anchored on chromosomes | 99.06% | 99.17% | 95.94% | 97.23% | 98.65% | 92.93% | 93.13% |
| Number of Ns (%) (Mb) | 0.04 (0.01%) | 20.9 (4.30%) | 0.18 (0.04%) | 3.7 (0.70%) | 0.08 (0.02%) | 0.05 (0.01%) | 0.05 (0.01%) |
| Number of genes | 34,451 | 35,669 | 35,669 | 35,075 | 33,662 | 35,986 | 36,439 |
| Busco Complete (%) | 1592 (98.7%) | 1581 (98.0%) | 1592 (98.7%) | 1593 (98.7%) | 1547 (95.8%) | 1596 (98.9%) | 1594 (98.8%) |
| Cumulative chromosome assembly size (Mb) | 535.9 | 468.6 | 477.4 | 509.8 | 531.1 | 475.4 | 490.3 |
| %Repeat (includes tandem repeat, satellites, Low complexity and telomers) | 62.13% | 53.59% | 57.13% | 60.82% | 62.18% | 58.03% | 58.88% |
| Retroelements (%) | 48.41% | 40.81% | 42.64% | 46.21% | 47.83% | 44.21% | 45.14% |
| DNA transposons (%) | 12.03% | 11.31% | 12.81% | 13.09% | 13.10% | 12.33% | 12.18% |
| BioProject | PRJEB72060 | PRJEB72058 | PRJEB72059 | PRJEB26661 | PRJEB72062 | PRJEB72061 | PRJEB72061 |

* Number of fasta sequences in the assembly.

Genetic mapping data were used to anchor the scaffolds and validate our assemblies. As a result, all six are chromosome-scale assemblies (Table 1, Supplementary Method 1, and Supplementary Note 1) and reach the Earth BioGenome Project recommendations[32], with a high proportion of each assembly anchored on chromosomes (96.6% on average), and Busco completeness ranging from 95,8% to 98,9% (Table 1). Around 35,000 gene models were predicted in each assembly, ranging from 33,662 genes in *M. textilis* assembly to 36,439 genes in Pisang Madu.

These assemblies were then used for comparative genome and phylogenetic analyses alongside four previously published Musaceae genome assemblies: *M. a.* ssp. *malaccensis* reference sequence[20], *M. a.* ssp. *malaccensis* haplotype assemblies[24], *M. balbisiana*[22] and *Ensete glaucum*[26] as outgroup.

**The Pisang Madu cultivar assembly provides access to the genomes of two unknown ancestors**
The Pisang Madu cultivar genome was previously shown to involve several ancestries including two unknown ancestral contributors[8]. In silico chromosome ancestry painting of the Pisang Madu assembly revealed that a full chromosome set (Pisang Madu H1) originated from one of these unknown ancestors (Fig. 2a). Based on a global genome comparison with available sequencing data Martin et al.[8] suggested that one of the unknown contributors of Pisang Madu could correspond to *M. a.* ssp. *halabanensis*. Here, we confirm this hypothesis by comparing haplotype sequences of two regions, i.e. alcohol dehydrogenase (*ADH*) and granule-bound starch synthase (*GBSS*), which are shared in *M. a.* ssp. *halabanensis* populations from Indonesia[33], to homologous regions of *Musa* genome assemblies. Phylogenetic analyses showed complete sequence identity or very close relationship between *M. a.* ssp. *halabanensis* and Pisang Madu H1 for the two regions (Fig. 2c, d). These findings support the *M. a.* ssp. *halabanensis* origin of Pisang Madu H1 (hereafter referred to as *M. a. halabanensis* assembly). The second haplotype of Pisang Madu (H2) consisted of a mosaic involving four ancestries, including the second unknown ancestry with large regions that together represent at least 30% of this haplotype (Fig. 2b). This highlights the importance of the Pisang Madu cultivar and its unique genomic ancestry in understanding the complex evolutionary history of cultivated bananas.

**Uncovering the phylogenetic position of unknown ancestors**
We identified 10,635 orthologous genes (out of 25,490) from the Pisang Madu H2 assembly regions corresponding to the unknown ancestor. These genes were used to perform phylogenetic analysis and estimate divergence times. The resulting phylogenetic tree, shown in Fig. 3, indicated that all *M. acuminata* subspecies, *M. schizocarpa*, and the unknown ancestor form a monophyletic group that diverged from *M. balbisiana* approximately 4.4 Mya ago. In this group, three well-supported monophyletic subgroups were observed: (i) a basal subgroup involving *M. schizocarpa* and *M. a.* ssp. *halabanensis* (Supplementary Fig. 1), (ii) a subgroup involving *M. a.* ssp. *zebrina* and the unknown ancestor (Supplementary Fig. 1g-h and Supplementary Fig. 2), and (iii) as expected, a clade containing the three *M. a.* ssp. *malaccensis* assemblies. These analyses demonstrate the close relationship between the unknown ancestor and *M. a.* ssp. *zebrina* and between *M. a.* ssp. *halabanensis* and *M. schizocarpa*. The divergence time of all these species from *M. textilis* was estimated at 7.8 million years ago (Mya).

**Tracing back the emergence of reciprocal translocations and refining their structure**
Global genome comparisons revealed general synteny conservation within *M. acuminata* assemblies and with *M. schizocarpa* and *M. balbisiana* assemblies, with a few exceptions (Fig. 3). These exceptions included some inversions observed in (peri)centromeric regions, as well as large reciprocal translocations and inversions that were previously reported in different *Musa* lineages[13,14,22,24,34,35].

Our comparison confirmed these rearrangements and refined their structure (Fig. 4, Supplementary Fig. 3, Supplementary Fig. 4). The *M. a.* ssp. *malaccensis* haplotype assemblies (h1, h2) showed that the chr1/chr4 rearrangement previously found in part of the *M. a.* ssp. *malaccensis* gene pool involved a reciprocal translocation as well as an inversion of the translocated fragment from chromosome 1 (Fig. 4a). The *M. a.* ssp. *zebrina* assembly showed that the chr3/chr8 rearrangement previously found resulted from a reciprocal translocation involving extremities of chromosomes 3 and 8 and an inversion of only a large part of the translocated fragment originating from chromosome 3 (Fig. 4b). Finally, the Pisang Madu H2 assembly showed that the chr1/chr7 rearrangement was complex and could tentatively be

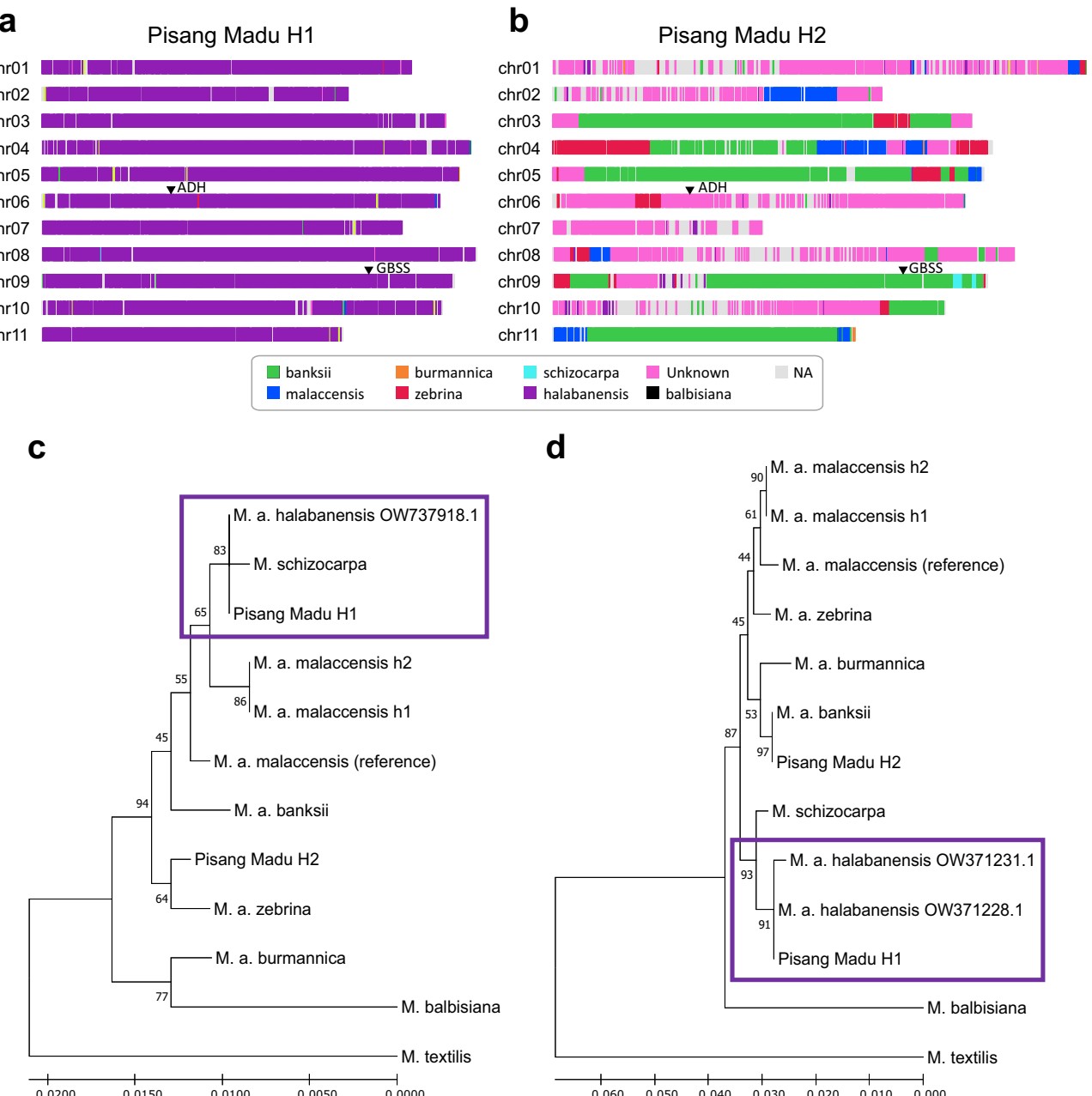

**Fig. 2 | In silico chromosome ancestry painting of the Pisang Madu assembly and phylogenetic analysis.** In silico chromosome ancestry painting of the haplotype 1 **a** and haplotype 2 **b** assembly of Pisang Madu. Colour codes indicate the ancestors involved, as defined by Martin et al.[8]. 'Unknown' is for the remaining unknown ancestry and 'NA' corresponds to regions in which no origin could be attributed. Maximum likelihood phylogenetic analysis (GTR + Γ substitution model) of *ADH* **c** and *GBSS* **d** haplotypes. The trees were obtained with MEGAX; branch lengths are measured according to the number of substitutions per site, and the percentage of trees in which the associated taxa clustered together is shown alongside the branches. Sequences that are grouped in polytomy together with *M. a.* ssp. *halabanensis* are enframed in purple in **c** and **d**. Black arrows locate the *ADH* (chr06) and *GBSS* (chr09) on each Pisang Madu haplotypes assemblies. Source data are provided as a Source Data file.

explained by three major events: an inversion within chromosome 7 followed by a reciprocal translocation between chromosomes 1 and 7, and an inversion within the resulting chromosome 1T7 (Fig. 4c, Supplementary Fig. 4). This rearrangement resulted in a small acrocentric chromosome 7 (chr7T1) and a large chromosome 1 (chr1T7) with two repeat-rich regions typical of (peri)centromeric regions, i.e. one at the extremity of the chromosome and one framed by two gene-rich regions (Fig. 4c). Moreover, the in silico chromosome ancestry painting of Pisang Madu H2 showed that the breakpoints of this rearrangement were located in regions corresponding to the unknown ancestor (Supplementary Fig. 5).

As expected, the *M. textilis* assembly was organized into 10 chromosomes. Its comparison with other *Musa* assemblies revealed globally conserved structures for chromosomes 3, 4, 6 and 10, with large chromosomal rearrangements affecting the remaining chromosomes (Fig. 3 and Supplementary Fig. 6-8). These structural events involved the reshuffling of segments from 2 to 4 different chromosomes, resulting in a basic chromosome number of x = 10 compared to x = 11 in other *Musa* assemblies.

Phylogenetic analysis allowed us to place the rearrangements within the evolutionary context of this species complex. The ancestral chromosome structure within the *M. acuminata*/*M. schizocarpa*

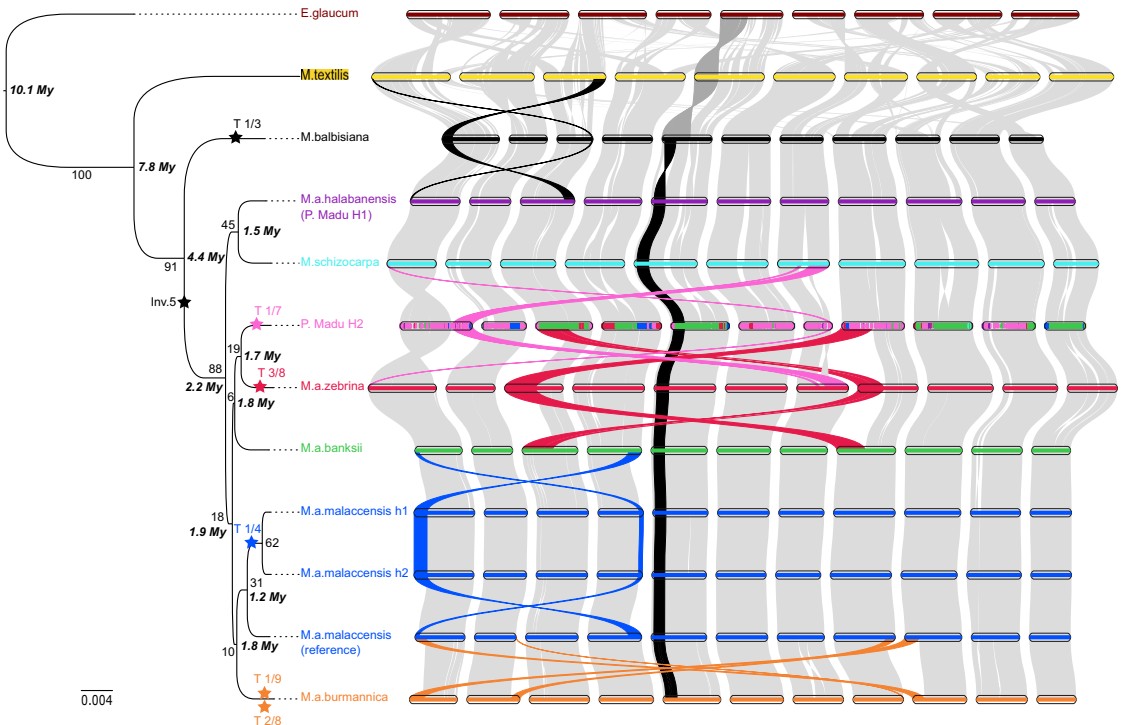

**Fig. 3 | Global synteny comparison and phylogenetic analysis of the assemblies.** Assemblies are colored based on their origin. Syntenic regions are indicated by gray ribbons, while colored ribbons indicate translocated regions. Translocation events are indicated by stars on the maximum likelihood phylogenetic tree. This tree was constructed using only the genes in regions assigned to the unknown ancestor in the P. Madu H2 assembly and their orthologs in the other assemblies. Branch support, calculated as the proportion of CDS gene phylogenies supporting the branch, are indicated. Divergence times, in million years (My), are indicated in bold italics close to nodes. Source data are provided as a Source Data file.

lineage was found to correspond to that of the *M. a.* ssp. *malaccensis* reference, *M. a.* ssp. *banksii*, *M. a.* ssp. *halabanensis*, and *M. schizocarpa* assemblies. It evolved through reciprocal translocations that emerged within 2.2 Mya in the lineages of *M. a.* ssp. *malaccensis* (chr1/chr4 translocation), *M. a.* ssp. unknown (chr1/chr7 translocation), *M. a.* ssp. *zebrina* (chr3/chr8 translocation*)* and *M. a.* ssp. *burmannica* (chr1/chr9 and chr2/chr8 translocations). The chr7/chr8 translocation described by Martin et al.[14] was not represented in the current assemblies but also emerged within the *M. a.* ssp. *burmannica* lineage. A chr1/chr3 reciprocal translocation emerged in the *M. balbisiana* lineage and a large inversion on chromosome 5 occurred in the *M. acuminata/M. schizocarpa* lineage. This suggested that the ancestral genome structure at the basis of the *M. acuminata/schizocarpa/balbisiana* phylogenetic group corresponded to the *M. acuminata/M. schizocarpa* ancestral chromosome structure but without the inversion on chromosome 5.

### The distribution of repeated sequences reveals variations in the (peri)-centromeric regions among *Musa*

Analyses of the repeated fraction of the *Musa* and *Ensete* genomes showed that the proportion of repeats ranged from around 53% up to 62% across the genomes, with *M. balbisiana* and *M. a.* ssp. *banksii* assemblies being in the lowest range and *M. textilis* and *M. a.* ssp. *zebrina* assemblies being in the highest (Table 1, Supplementary Data 1). Copia LTR retrotransposons represented the most abundant TE superfamily in the *M. balbisiana*, *M. schizocarpa* and all *M. acuminata* assemblies, with the SIRE/Maximus family being the most represented (18-24% of the genome assemblies). As previously observed, large clusters of tandem repeats (CL18 and CL33[36] and TR01) were found in most of the *Musa* assemblies (Fig. 5a,b and Supplementary Fig. 9-21).

Retrotransposons from the SIRE/Maximus family were found to be particularly abundant in peri-centromeric regions (Fig. 5a,b, Supplementary Fig. 9-21). The Nanica long-interspersed element (LINE), and

CRM retrotransposons, previously found in all centromeric regions of *M. a.* ssp. *malaccensis*[20], *M. schizocarpa* and *M. balbisiana* assemblies[19,22,23], were also found in the *M. acuminata* assemblies, as well as in *M. textilis*. These landmarks of centromeric and pericentromeric regions showed that *M. schizocarpa*, *M. balbisiana* and *M. acuminata* subspecies genomes were all mainly organized in meta-centric chromosomes, with the exception of chromosomes 1, 2 and 10 which are acrocentric (Fig. 5a,b, Supplementary Fig. 9-21). Chromosome 1 of the Pisang Madu H2 assembly and chromosome 5 of the *M. textilis* assembly showed an unusual pattern with two separated SIRE/Maximus rich regions (Supplementary Figs. 12 and 17). Since the *M. textilis* chromosome 5 structure was supported by genetic mapping data (Supplementary Fig. 22), it may reflect a relatively recent inversion (Fig. 3 and Supplementary Figs. 6–8). Similarly, we found that the unusual structure of Pisang Madu H2 chromosome 1 was the consequence of a recent rearrangement.

Typical major 45S and 5S ribosomal DNA (rDNA) clusters were observed on one chromosome arm in the assemblies, with occasional additional sites found on other chromosome arms. More surprisingly, 45S rDNA and 5S rDNA clusters were observed together with Nanica/CRM clusters in the centromeric region of all chromosomes in *M. schizocarpa* and some chromosomes in *M. a.* ssp. *halabanensis* (Fig. 5b and Supplementary Fig. 11), with varying size. Cytogenetic analysis confirmed the presence of 45S and 5S rDNA clusters together with Nanica clusters in the centromeres of *M. schizocarpa* (Fig. 5c–e). Additionally, 5S rDNA clusters were present in the centromeric region of chromosomes 1 and 3 in *M. a.* ssp. *malaccensis* (Supplementary Figs. 13–15).

### Structural insights from ancestral contributors to current Cavendish assemblies

Two triploid chromosome-scale genome assemblies of the Cavendish cultivar were recently released[29,31]. The genome of this cultivar was

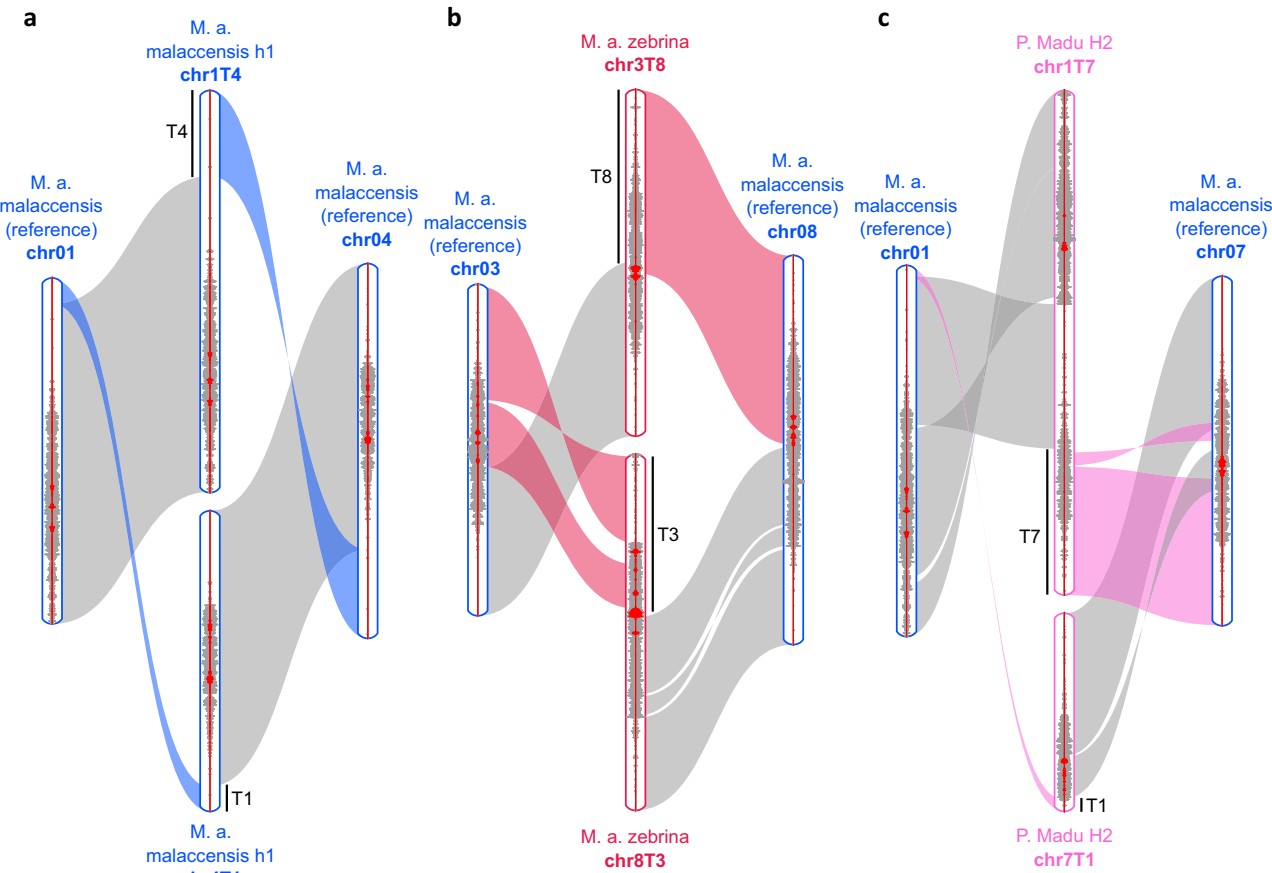

**Fig. 4 | Structure of the 1/4, 3/8 and 1/7 chromosome rearrangements compared to the *M. a. malaccensis* reference structure.** The reference structure corresponds to the *M. a.* ssp. *malaccensis* DH-Pahang (M. a. malaccensis (reference)) assembly[20] compared to **a** 1/4 rearranged structure of the *M. a.* ssp. *malaccensis* h1 (M. a. malaccensis h1 chr1T4 and chr4T1) assembly[24]; **b** 3/8 rearranged structure of the *M. a.* ssp. *zebrina* assembly (M. a. zebrina chr3T8 and chr8T3); **c** 1/7 rearranged structure of the Pisang Madu haplotype 2 assembly (P. Madu H2 chr1T7 and chr7T1). Syntenic regions are indicated with ribbons. Coloured ribbons correspond to translocated fragments. Chromosome and translocated fragments were named according to the nomenclature of Martin et al.[14]. Curves within the chromosomes correspond to (peri)centromeric repeats in grey and Nanica in red. Source data are provided as a Source Data file.

shown to consist in a complex mosaic resulting from recombination between several ancestral contributors: *M. a.* ssp. *banksii*, *M. a.* ssp. *zebrina*, *M. a.* ssp. *malaccensis*, the unknown contributor, *M. schizocarpa*, and possibly *M. a.* ssp. *halabanensis*[8]. Applied to the Huang et al.[29] assembly, in silico ancestral chromosome painting allowed us to estimate these contributions to 26%, 24%, 20%, 11%, 1%, and 1% respectively (Fig. 6). In the Li et al.[31] assembly, these contributions were only partially represented since we show that homologous haplotypes (originating from the same subspecies) have been collapsed or stacked, resulting in homologous chromosomes of very variable sizes (Supplementary Figs. 23–33). In addition, chromosome segments contributed by the unknown ancestor were not differentiated from the ones contributed by *M. a.* ssp. *zebrina* in this assembly. One such segment includes a RLP (receptor-like protein) gene cluster, suspected to be involved in *Fusarium oxysporum* f.sp. *cubense* (Foc) Race1 resistance[31], which we have shown here to be contributed by the unknown ancestor (Supplementary Fig. 32).

The Cavendish genome was suggested to contain three large reciprocal translocations (chr3/chr8, chr1/chr7 and chr1/chr4), in line with the reciprocal translocations present in its ancestral contributors[14]. However, none of these rearrangements were found in the Cavendish assembly of Li et al.[31] and they were only partially represented in the one of Huang et al.[29] (Fig. 6d, Supplementary Figs. 34–36). In the Huang et al.[29] assembly, some of the translocation breakpoints were represented (on chr08-h2, chr04-h2, chr01-h3 and chr07-h2) but the remaining parts of chromosomes involved in

these translocations did not show the expected rearranged structures (Fig. 6d). We performed BAC-FISH analysis on Cavendish chromosomes using BAC pairs that were located on distinct chromosomes in the reference structures but on the same chromosome in the translocated structures (Fig. 6d). In all three instances, the BAC pairs were found together on a single chromosome, thereby confirming the presence of translocated chromosomes in the Cavendish genome, in contrast with the observed Cavendish assembly structure (Fig. 6d–f). The translocated structures were assembled in the ancestral genome assemblies; these assemblies could therefore be useful in the future to guide triploid cultivars assemblies.

Finally, we recently proposed that the Cavendish genome resulted from an un-recombined 2x gamete from the diploid Mchare cultivar and a 1x gamete from a close relative of the diploid Pisang Madu cultivar[37]. We painted chromosomes of the Cavendish assemblies, according to the contributing gametes and found that around half and nearly all were recombinants between gamete haplotypes in the Huang et al.[29] and Li et al.[31] assemblies, respectively, (Fig. 6c and Supplementary Figs. 23–33), hence suggesting chimerism in the haplotype assemblies.

## Discussion
Banana is largely grown as a monoculture, which renders it vulnerable to emerging biotic stresses, such as Fusarium TR4, a fungus that is currently spreading and devastating banana crops worldwide[38] or

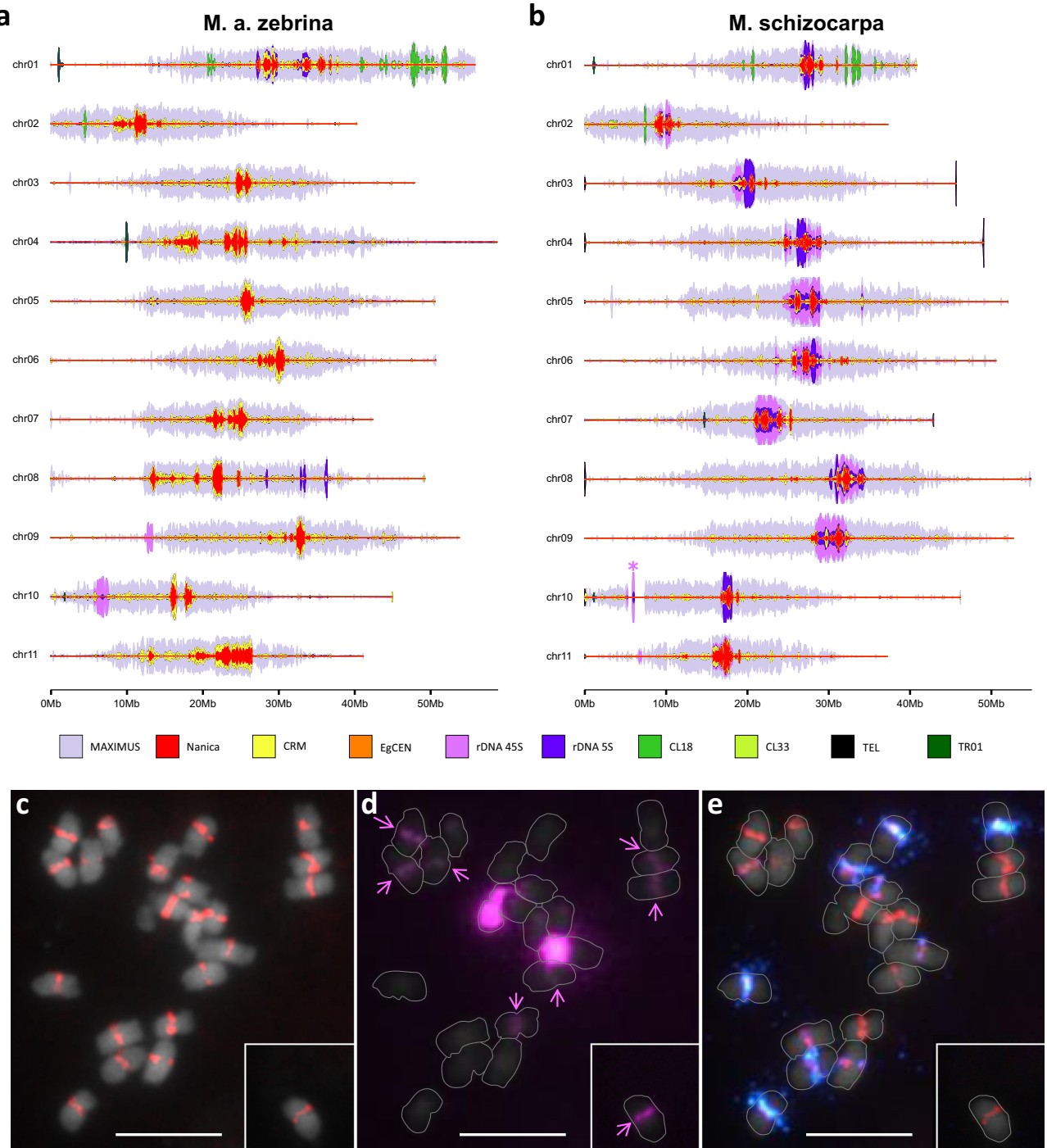

**Fig. 5 | Transposable elements and tandem repeats.** Stacked curves representing the density of SIRE/Maximus transposable elements, centromeric sequences and tandem repeats along (**a**) the *M. a.* ssp. *zebrina* and (**b**) *M. schizocarpa* assemblies. The pink asterisk indicates that the region contains a stretch of N impacting the full representation of the rDNA cluster. Typical results of fluorescence in situ hybridization (FISH) on metaphase chromosomes of *M. schizocarpa* with (**c**) Nanica transposable elements detected in red, (**d**) 45S rDNA detected in pink and (**e**) Nanica transposable element detected in red and 5S rDNA detected in blue. White horizontal bars in (**c**), (**d**) and (**e**) represent 5.1 µm. Arrows in (**d**) indicate 45S rDNA detected with lower signal intensity. The inserts represent a chromosome that was offset from the presented field of view. Source data are provided as a Source Data file.

Black sigatoka disease, caused by another fungus, which requires massive pesticide treatments[39]. There is thus an urgent need for breeding disease-resistant bananas, but breeding strategies have been hampered by the sterility or very low fertility of cultivars and scant knowledge available until recently on cultivar genome architecture and on the genetic determinism of the agronomic traits.

The genomes of banana cultivars were recently shown to consist in complex mosaics involving ancestral contributors from mainly nine wild genetic groups, including two contributors of unknown origin[8]. In this study, we produced chromosome-scale genome assemblies representing five of these genetic groups (*M. a.* ssp. *zebrina*, *M. a.* ssp. *burmannica*, *M. a.* ssp. *banksii*, *M. schizocarpa*, and *M. textilis*). We also produced a chromosome-scale genome assembly of one of the previously unknown contributors, which we confirmed to be *M. a.* ssp. *halabanensis*, and an assembly that partially corresponded to the second yet unknown

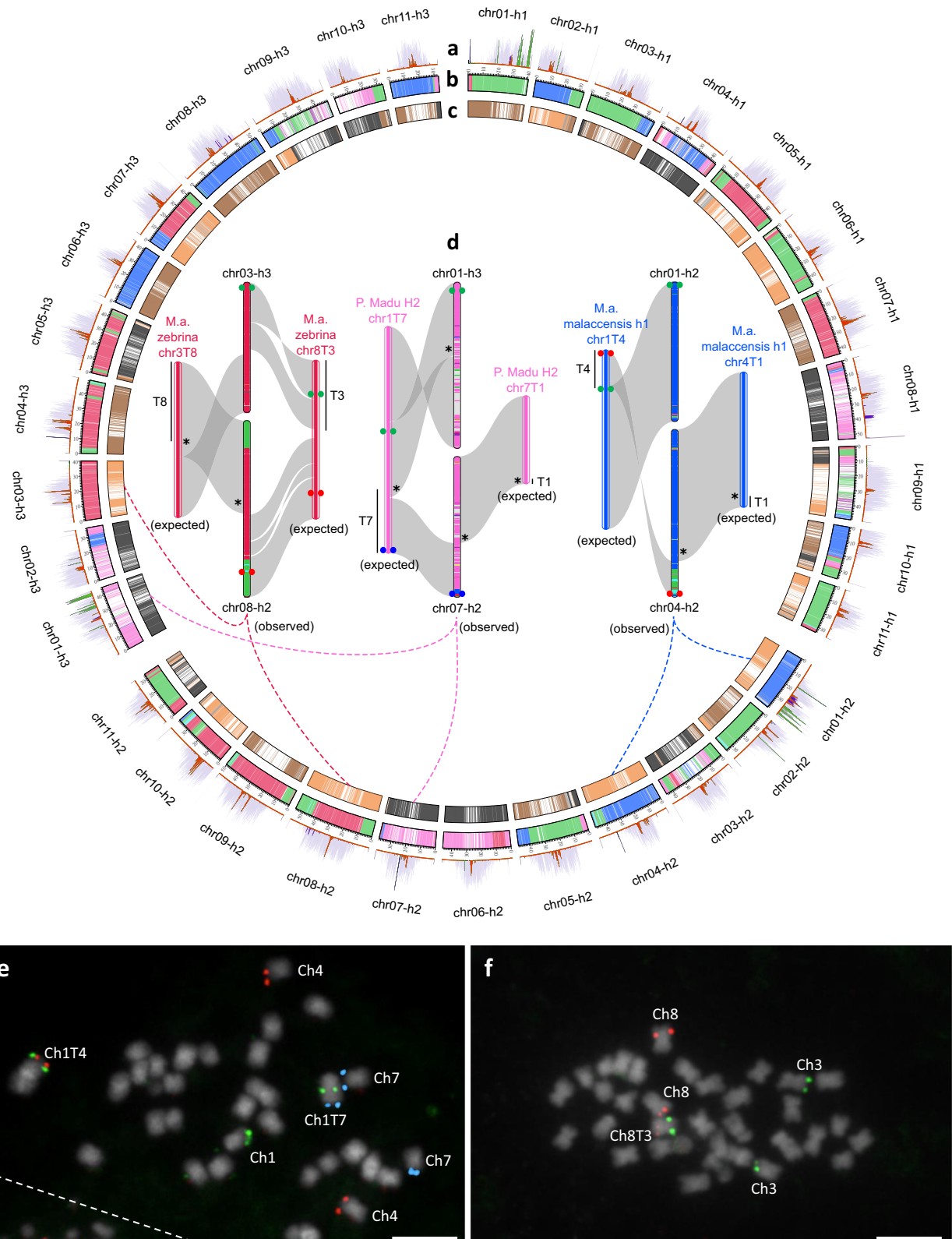

ancestor. These two last assemblies represent particularly essential resources since no pure representative of these contributors to banana cultivars are available in ex situ collections. These seven assembled chromosome-scale genomes, together with *M. a.* ssp. *malaccensis*[20] and *M. balbisiana*[22] genome assemblies complete the set of reference genomes for all the main contributors to cultivated bananas.

We showed that despite the advance of sequencing and assembly technologies, it is still difficult to achieve accurate assembly of triploid cultivars genomes due to their complex ancestral mosaic architecture. It is thus important to take into account information on their ancestral mosaic structure obtained through in silico chromosome ancestry painting[8] and when possible on their parents[37] to adapt the assembly strategy. In particular, because their chromosomes derived from

**Fig. 6 | In silico chromosome ancestry painting of a triploid Cavendish cultivar genome assembly and comparison with ancestral contributor assemblies.** Circular representation of the Cavendish haplotype assembly (Baxijiao accession[29]) with: **a** the distribution of tandem repeats and SIRE/Maximus transposable elements along chromosomes (color code from Fig. 5), **b** In silico chromosome painting according to the ancestral contributors (color code from Fig. 2), **c** In silico chromosome painting according to suggested parental gametes[37] with black for the 1x gamete and orange and brown for the two haplotypes of the 2x gamete. **d** Comparison of the assembled Cavendish chromosomes that should have the 1/4, 1/7 and 3/8 rearrangements on the basis of their ancestral origin, with the reference rearranged structures found in ancestral contributor assemblies. The reference structure used for the 1/4 reciprocal translocation corresponds to *M. a.* ssp. *malaccensis*[24] (M. a. malaccensis h1 chr1T4 and chr4T1), that used for the 1/7

reciprocal translocation corresponds to Pisang Madu H2 (P. Madu H2 chr1T7 and chr7T1) and that used for the 3/8 reciprocal translocation corresponds to *M. a.* ssp. *zebrina* (M. a. zebrina chr3T8 and chr8T3). Dashed lines indicate chromosomes from the Cavendish assembly that were compared. Translocated segments on the reference translocated structure are indicated by vertical black bars. Gray shapes locate the syntenic regions. Stars indicate the translocation breakpoints that are present in the assembly. Green, red and blue dots on **d** locate the BACs used for BAC-FISH analysis on metaphase chromosomes of Cavendish (Grande Naine accession) to validate the presence of the 1/4 and 1/7 **e** and 3/8 **f** translocated structures in Cavendish. White horizontal bars in **e** and **f** represent 5.1 μm. White dashed line separated chromosomes from another cell. Source data are provided as a Source Data file.

recombination between various ancestral contributors with distinct chromosome architecture[8,14], taking into account the origin and the expected structure of each haplotype during the assembly process, instead of relying on unique reference genome as a guide is important. The reference genome assemblies of ancestral contributors produced in this study will be particularly useful in this respect. Furthermore, accurate genome assemblies and knowledge of the ancestral genome architecture of successful cultivars, such as Cavendish, are invaluable for reconstructing breeding strategies. This can help produce hybrids with a similar genomic makeup but with the introgression of disease resistance genes.

These genome assemblies are also crucial for identifying favourable genes and alleles involved in QTLs for agronomic traits[40–43]. They aid in pinpointing the genetic groups from which these favourable alleles originated, thereby guiding germplasm selection for breeding programs based on targeted traits. Finally, precise knowledge on large chromosome structural variations and their impacts on chromosome recombination is essential for designing strategies to exploit these alleles in breeding programs.

The still unknown ancestor seems of particular interest since it is present in many successful dessert bananas including Cavendish and Gros Michel[8]. Biabiany et al.[42] have shown that the translocated chromosome 1T7 in Pisang Madu was associated with a QTL for pulp acidity. As this translocated chromosome originates from the unknown ancestor, it showed that this unknown ancestor contributed an important dessert banana fruit quality trait. This unknown ancestor also brought to Cavendish a RLP cluster that was suggested to be involved in its Fusarium Foc R1 resistance and initially attributed to *M. a.* ssp. *zebrina*[31]. Because of its Foc R1 resistance Cavendish replaced the previously dominant Gros Michel cultivar that was decimated by the disease in the 50th.

The unknown contributor is involved in cultivars collected in an area spanning from Thailand to Papua New Guinea[8]. In the phylogenetic tree, it clusters close to *M. a.* ssp. *zebrina*, a taxa that is found in the island of Java, in Indonesia. Since the organisation of the tree mostly correlates with the geographic distribution of *M. acuminata* sub-species, this unknown contributor may originate from Southeast Asian islands rather than the Malayan peninsula or New Guinea island. This information together with the assembled part of its genome we produced, will help identify pure accessions from this genetic group and better target geographic areas to explore.

The complicated geological and climatic history in Southeast Asia contributed to the high species richness in the region[1] and shaped the diversity of the *Musa* genus[10]. Our analyses suggest two mechanisms for chromosome evolution within the represented *Musa* (sub)species i.e. large chromosome rearrangements, including reciprocal translocations and inversions, and centromere differentiation through integration of different types of repeated sequences.

Large chromosome rearrangements were reported as drivers of speciation in various organisms[44–46]. Within *Musa*, ancestral chromosome rearrangements resulted in different basic chromosome

numbers between the clade of *M. textilis (sect. Callimusa)* and the clade of *M. balbisiana/M. schizocarpa/M. acuminata (sect. Musa)* (this work and[27,28]). In addition, several large chromosome rearrangements have previously been reported within the clade of *M. balbisiana/M. schizocarpa/M. acuminata*[13,14,34,35,47,48]. Our genome assemblies confirmed the presence of seven large reciprocal translocations and one large inversion in the *M. acuminata/schizocarpa/balbisiana* group and showed that they occurred in different phylogenetic branches within 4,4 My. Three of the reciprocal translocations were found with more complex structures than initially proposed[14,34], i.e. involving inversions. Interestingly, these inversions in structurally heterozygous individuals can induce improper segregation of chromosomes at meiosis leading to reduced fertility, thus explaining the absence of recombination that was observed in the corresponding regions[14]. Furthermore, reciprocal translocations are in many cases preferentially transmitted[14], which may contribute to their fixation in populations. Moreover, part of the gametes from heterozygous individuals are aneuploid, thereby reducing the hybrid fitness and in turn reinforcing the speciation process[14].

Analysis of the *Musa* assemblies also revealed particular and diverse repeated content in the (peri)centromeric regions of chromosomes. Centromeres are essential regions of eukaryotic chromosomes that mediate kinetochore assembly and microtubule spindle attachment, allowing proper segregation of chromosomes during cell division. Centromeric regions are predominantly composed of repeated sequences, mainly short tandem repeats (satellites) and/or retrotransposons, such as CRM in most plants[49]. Despite their conserved function, their size, structure and repeat content vary markedly within and between species[50,51]. This variability was shown to be generated by cycles of transposons invasion and purging through satellite homogenization, a process which drives centromere evolution and ultimately contributes to speciation[49]. In current *Musa* assemblies, short tandem centromeric repeats typical of other plant species are not found. Instead, the centromeric regions of all chromosomes contain a LINE element named Nanica, along with CRM retrotransposons; a combination reported in only a few other plants[52]. In contrast, *Ensete glaucum* lacks the Nanica element and instead features a short tandem repeat in its centromeric regions[26]. Unexpectedly, we also found that in *M. a.* ssp. *halabanensis*, *M. schizocarpa* and to some extent *M. a.* ssp. *malaccensis*, other types of repeated sequences, namely rDNA 5S and 45S, were found in centromeric regions of chromosomes. Ribosomal DNA (rDNA) codes for the rRNAs used in the production of ribosomes; their function is to synthesise proteins by decoding the information contained in messenger RNA. In most eukaryotes, including *Musa*, rDNA consists of tandemly repeated arrays of a few genes located in the nucleolus organizer region (NOR) of one or sometimes a few chromosome arms[53]. This is, to our knowledge, the only report of such rDNA sequence integration in centromeric regions of plants. The functional region of the centromere is defined by loading of a specific histone 3 variant (*CENH3*), it would thus be interesting to test whether the Nanica LINE and rDNA sequences that we detected in *Musa* centromeric regions, can be sites of *CENH3* loading. Yet, these findings of

various types of centromeric repeats among *Musa* sub-species and close relatives provide an interesting model to study the mechanisms involved in the cyclic invasion of various types of repeated sequences and purging through satellite homogenization.

The *Musa* genus originated in Northern Indo-Burma and *Musa* species dispersed and evolved throughout Southeast Asia and New Guinea, in a context of land connection-disconnection events and climatic fluctuations[10]. The current geographical distribution of the *M. acuminata* subspecies and of *M. schizocarpa* reflects this dispersion and diversification process (Fig. 1). The phylogeny grouped most *M. acuminata* subspecies together as expected but raised questions about classification and evolution, with the grouping of *M. a.* ssp. *halabanensis* and *M. schizocarpa* in a monophyletic group, basal to other *M. acuminata* subspecies. This grouping is consistent with the presence of 45S and 5S rDNA clusters in a large proportion of the chromosomes of *M. a.* ssp. *halabanensis* and *M. schizocarpa*. The classification of *M. schizocarpa* as a species was based on morphological characters, notably large seeds and self-peeling fruits at maturity[54,55]. It is interesting to note that the latter characteristic, which is rare in the Musaceae, was also noted in *M. a.* ssp. *halabanensis*[33,56]. In addition, *M. a.* ssp. *halabanensis* was considered by Meijer[56] as a separate species but was later included as subspecies to the *M. acuminata* by Nasution et al.[57]. Geographically, *M. a.* ssp. *halabanensis* is only found along the western side of Sumatra island, in western Indonesia[33], whereas *M. schizocarpa* was only described so far on the New Guinea island[54,55]. Historical changes in land connectivity and climate can explain the apparent discrepancy between their close relatedness and distant localisation. In Papua New Guinea, *M. schizocarpa* is sympatric with another *M. acuminata* subspecies namely *M. a.* ssp. *banksii*. They often grow side-by-side and hybridize. However, their hybrids have very low fertility and both taxa remain distinct[55,58]. Both taxa also displayed chromosomes with the ancestral structure, so do not differ by large chromosome rearrangements. The integration of large 45S rDNA and 5S rDNA clusters into *M. schizocarpa* centromeres may act as a reproductive barrier between these taxa.

In the Indo-Australian archipelago, the cyclical succession of glacial and warm periods in the Pleistocene and associated changes in land connectivity and vegetation cover could have promoted speciation in *Musa* notably through vicariance[59]. Within the *M. acuminata/M. schizocarpa* phylogenetic group, *M. a.* ssp. *halabanensis* and *M. schizocarpa* form a basal subgroup that may have diverged early from the other *M. acuminata* subspecies through the differentiation of their centromeres. The other *M. acuminata* subspecies form another subgroup within which the divergence involved large chromosome rearrangements that were each found specific to one subspecies. Our analysis of these genome assemblies suggests that the speciation process in *Musa* involved distinct genomic drivers such as large chromosome rearrangements and centromeres differentiations. Therefore, genome assemblies from this rich *Musa* (sub)species complex represent important resources for studying the genomic drivers of speciation.

## Methods

### Plant materials and DNA extraction

The PT-BA-00024 accession from *M. a.* ssp. *banksii*, the PT-BA-00228 accession from *M. textilis* and the *Musa* diploid hybrid (Pisang Madu - PT-BA-00304) were used to produce the *M. a.* ssp. *banksii*, *M. textilis*, and Pisang Madu assemblies, respectively. These plant materials were obtained from the CIRAD-INRAE Biological Resource Centre for Tropical Plants (CRB-PT) in the West Indies (Guadeloupe, France). *M. schizocarpa* (https://doi.org/10.18730/9KW4W, ITC926) used to produce the *M. schizocarpa* assembly was obtained from the Bioversity International Transit Center in Leuven (Belgium). To increase homozygosity, self-crosses of Maia'Oa (PT-BA-00182), and Calcutta 4 (PT-BA-00051) accessions were performed at the CIRAD banana breeding

platform in Guadeloupe and one progeny from each cross was selected to produce the *M. a.* ssp. *zebrina* and *M. a.* ssp. *burmanica* assemblies. DNA extractions were performed using MATAB procedure[20].

### Sequencing data and chromosome assembly

Two primary sequencing techniques were used to produce the *M. a.* ssp. *banksii*, *M. a.* ssp. *zebrina*, *M. a.* ssp. *burmannica* and *M. textilis* assemblies: Illumina and Oxford Nanopore Technologies (ONT). For Illumina sequencing, a PCR-free library was prepared for each sample and genomic DNA was sequenced with Illumina HiSeq 2500. Quality control involved trimming of low-quality nucleotides and removal of adapters[60]. For ONT sequencing, libraries were prepared using various kits (SQK-LSK108, LSK-SQK109) and sequenced on MinION or PromethION R9.4 flow cells. Different approaches were used for fragmenting and selecting DNA sizes depending on the *Musa* species. Multiple assemblers such as Necat[61], Flye[62], Raven[63], SMARTdenovo[64], and Redbean[65] were used and compared. For each species, the best assembly (based on the contiguity and cumulative size) was selected and further polished using Racon[66], Pilon[67] and/or Hapo-G[68]. Finally, optical maps and/or Hi-C sequencing data were used for scaffolding. In contrast, Pisang Madu was sequenced using the Pacific Biosciences HiFi technology and assembled using Hifiasm[69] in order to obtain the sequences of the two haplotypes.

Unanchored scaffolds from assemblies were classified through an additional procedure, excluding chloroplastic scaffolds but including mitochondrial derived scaffolds based on BLAST similarity, and grouping of the remaining scaffolds (with <99% identity) into a chrUnrandom sequence. All of these assemblies were validated using Merqury and genetic map anchoring.

*M. schizocarpa* assembly was improved to obtain version 2 without generating new data, but instead through the integration of optical maps and the manual incorporation of identified missing contigs.

The details of these procedures are extensively described in the Supplementary Methods 1 and Supplementary Note 1 (Supplementary Fig. 37– 40, Supplementary Data 2–7).

### Pisang Madu haplotype parsing

The highly heterozygous diploid Pisang Madu genome was sequenced using the PACBIO HiFi technology to resolve both haplotypes. However, the parsing of contigs into haplotypes via the Hifiasm program was not optimal. To improve this parsing, we exploited genetic mapping to access haplotypes and parents-child trios so as to densify the phased markers. Phased sequence information was used to generate tags specific to each haplotype. Tags were used to parse contigs into haplotypes. The overall procedure is described in detail in Supplementary Fig. 41 and in Supplementary Methods 2 (Supplementary Figs. 42,43, Supplementary Data 8).

Contigs parsed to each haplotype were validated and anchored as described for other genomes. Due to structural heterozygosity involving chromosomes 1 and 7, recombination is blocked on regions of these chromosomes[14], thereby preventing ordering with a genetic map in these regions. The contigs in these regions were therefore ordered using DH-Pahang v4 as a guide. Contig junctions were then checked for the presence of nanopore reads overlapping these junctions.

Contigs that were unattributed to a haplotype were attributed to both haplotype assemblies following the procedure described for the other assemblies.

### In silico chromosome ancestry painting of assemblies

We developed a methodology to "paint" chromosomes of assemblies according to ancestral genetic groups. This was done by generating tags specific to each ancestral origin. These tags could then be aligned along chromosome assemblies and, based on the pattern of these alignments, an origin could be attributed to regions along

chromosomes. This process is described in the Supplementary Methods 3 (Supplementary Fig. 44, Supplementary Data 9) and summarized in Supplementary Fig. 45.

## Dot plot genomes comparison

Genomes were compared by mapping the *M. a.* ssp. *malaccensis* reference sequence[20] mRNA genes against assemblies using BLASTn (-evalue 1e-20 -out -num_threads 1 -max_target_seqs 1). The first best hits were selected and used to draw a dot-plot with a custom script.

## Genome synteny, phylogenetic analysis and divergence time

For theses analyses, the genome annotation of the *M. a.* ssp. *malaccensis* reference sequence (DH-Pahang v4)[20] was transferred to all genomes using liftoff v1.6.3[70] to ensure that the gene sets and annotations were homogenous among assemblies.

Orthologous genes and syntenic blocks were then searched using the jcvi v1.1.18 - MCscan tool[71]. Syntenic blocks were then drawn using jvci.

Phylogenetic analysis was performed using a subset of orthologous genes identified by jcvi tools. As MCscan analysis is a genome pairwise analysis procedure, syntenic blocks had to be associated between all assemblies. If more than 20 genes were shared between one DH-Pahang v4 syntenic block and a syntenic block of each other assembly, these blocks were identified as syntenic and shared genes between all of these blocks were considered as being orthologs. A total of 25,490 orthologous genes were identified.

As one of the assemblies (Pisang Madu H2) is hybrid with distinct origins including a large portion of the unknown ancestral origin, we further selected only genes identified in regions of the unknown ancestral origin of this assembly (Supplementary Data 10, Supplementary Fig. 46). This reduced the number of orthologous genes to 10,635. Each orthologous gene CDS was aligned using MAFFT v7.475[72] and aligned genes were concatenated to perform a global phylogenetic analysis using PHYMLv3.1[73] with the HKY85 model. In addition, a phylogenetic analysis was conducted for each orthologous gene alignments using PHYML v3.1 with the HKY85 model. The proportion of gene phylogenies supporting each branch of the global phylogeny was computed using the Phylo package (https://biopython.org/wiki/Phylo)[74]. The proportion of gene phylogenies supporting different positions of the *M. a.* ssp. *halabanensis* assembly and the unknown ancestries was also tested using the Phylo package.

In silico chromosome ancestry painting of other assemblies was performed to verify that they did not have large introgression segments from other origins (Supplementary Fig. 47).

The synonymous mutation (Ks) was calculated for each orthologous pair of the selected 10,635 gene CDS. Protein sequences were aligned with Clustal W[75] and PAL2NAL[76] was used to reconstruct the multiple codon alignment based on the corresponding aligned protein sequences. The Ks values were calculated with the Nei-Gojobori method implemented in PAML[77]. This process was performed using the synonymous_calc.py script (https://github.com/tanghaibao/biopipeline/blob/master/synonymous_calculation). Divergence times were estimated using the following formula

$$T = (\text{median Ks between two assemblies})/(2*4.5E - 9) \qquad (1)$$

where 4.5E-9 corresponds to the average synonymous substitution rate per year estimated in Musaceae[78]. Median Ks between two assemblies can be found in Supplementary Data 11.

## Validation of the *M. a.* ssp. *halabanensis* origin

The Pisang Madu H1 assembly origin was validated using sequences from two *GBSS* gene haplotypes (Genbank IDs: OW371228.1, OW371231.1) and one *ADH* gene haplotype (Genebank ID: OW737918.1) obtained from *M. a.* ssp. *halabanensis* from Indonesia[33]. Their homologous sequences in genome assemblies were identified using BLASTN[79]. *ADH* and *GBSS* haplotypes correspond to segments from the predicted genes *Macma4_06_g20230.1* (*ADH3*, chromosome 6) and *Macma4_09_g22760.1* (*GBSS1*, chromosome 9) respectively, in the DH-Pahang V4 reference genome[20]. All sequences were aligned using MAFFT-7.471[72]. Maximum likelihood phylogenetic analysis was performed using MEGAX[80] and the General Time Reversible + Gamma substitution model[81] (GTR + Γ substitution model). All positions containing gaps and missing data were eliminated (complete deletion option).

## Repeats in *Musaceae* assemblies

Repeat sequence catalogs were generated using EDTA v1.9.5[82] for each assembly and merged with Uclust[83] according to the 80/80 rule (two sequences are in a group if they share 80% identity for 80% of their length) to generate a global Musaceae repeated sequence set (22,958 non-redundant sequences) based on assemblies from *E. glaucum*, *M. textilis*, *M. beccarii*[27], *M. balbisiana*, *M. schizocarpa*, *M. acuminata* subspecies *(banksii, burmannica, malaccensis and zebrina)* and the Pisang Madu cultivar (halabanensis (H1) + H2 haplotypes). Similarly, LTR retrotransposons were specifically recovered in each assembly using LTR Harvest[84] and the results were filtered so as to only keep complete LTR retrotransposons (*i.e.* containing 80% of the consensus length of the Copia/Gypsy transposase protein domain). Sequences were compared to the plant intact LTR-RTs dataset of Zhou et al.[85] to identify LTR families and merged with Uclust according to the 80/80 rule to generate a global Musaceae LTR retrotransposons sequence set (4257 non-redundant sequences).

RepeatMasker v4.1.1 (http://www.repeatmasker.org) was used to mask repeats in assemblies with default parameters and -xsmall options with *Musaceae* repeats and LTR sets as custom libraries. LTR family abundance statistics were extracted from the RepeatMasker output files generated with Complete LTR retrotransposons set as custom library and overlapping hits were merged with the bedtools merge program[86].

Tandem repeats were recovered from assemblies using MREPS[87] with "-res 5 -minsize 500 -exp 40 -minperiod 100" option.

Musaceae centromeric sequences EgCEN[26] and Nanica[19], CL18 and CL33[36] and TR01 tandem repeats, DNA genic sequences of rDNA 5S, 5.8S, 16S and 26S extracted from the *Musa* DH Pahang reference sequence[20], and telomeric repeats were localized on assemblies using a blast procedure (https://github.com/institut-de-genomique/Pahang-associated-data)[20]. The sequence data are available on the Banana Genome Hub(http://banana-genome-hub.southgreen.fr).

## Molecular cytogenetics

Chromosome preparations and in situ hybridization were performed using classical methods for banana[88] excepted that roots have been treated for 6 hours, and probes labelled with direct Dyes. BAC clones (MAMB_51J24, MAMB_17B03, MAMB_37L22, MAMB_04C23, MAMH_04L23 and MAMH_51D16 obtained from the DH-Pahang accession[19] http://banana-genome-hub.southgreen.fr), Nanica, and 45S rDNA were labelled by random priming (Invitrogen; Thermo Fisher Scientific, Waltham, MA, USA) with Alexa 488-5-dUTP, Alexa 594-5-dUTP or CY3-dUTP. The 5S rDNA was labelled by PCR with CY3-dUTP. Chromosome preparations were incubated in RNAse A (100 ng μL⁻¹) and pepsin (100 mg mL⁻¹) in 0.01 M HCl Fluorescence images were captured using a cooled high-resolution black and white CCD camera (ORCA; Hamamatsu, Hamamatsu City, Japan) fitted on a DMRXA2 fluorescence microscope (Leica Microsystems, Wetzlar, Germany) and analysed using VOLOCITY (Quorumtechnologies Inc.).

## Reporting summary

Further information on research design is available in the Nature Portfolio Reporting Summary linked to this article.

## Data availability

The Illumina, ONT, PACBIO HiFi, Bionano Genomics, Hi-C data, assemblies and annotations generated in this study have been deposited in the European Nucleotide Archive under project PRJEB72282. The genome assemblies and gene and TE annotations are available at Banana Genome Hub [http://banana-genome-hub.southgreen.fr][89]. *GBSS* and *ADH* alignments, EDTA repeat database and LTR repeat database produced are available on the Banana Genome Hub [http://banana-genome-hub.southgreen.fr] in the download section and on CIRAD dataverse [https://doi.org/10.18167/DVN1/6TLMD3]. The GBS data are available in the Short Read Archive under the following projects PRJNA1078411, PRJNA1182927 and PRJNA667853. Germplasm is available at the CIRAD-INRAE Biological Resource Centre for Tropical Plants (CRB-PT) in the West Indies (Guadeloupe, France) for accessions PT-BA-00024 (*M. a.* ssp. *banksii*); PT-BA-00228 for *M. textilis*; PT-BA-00182 for *M. a.* ssp. *zebrina*; PT-BA-00051 for *M. a.* ssp. *burmannicoides* and PT-BA-00304 for the diploid cultivated Pisang Madu. Germplasm of *M. schizocarpa* is available at the Bioversity International Transit Center in Leuven (Belgium) under ITC0926 accession number. Publicly available sequencing data (PRJEB58004 [https://www.ncbi.nlm.nih.gov/bioproject/?term=PRJEB58004] and PRJEB26661) were also used in this study. Source data are provided with this paper.

## Code availability

Custom code has been added to vcfHunter toolbox that is available at Github [https://github.com/SouthGreenPlatform/VcfHunter].

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

## Acknowledgements

This research was supported by the Genoscope, the French Alternative Energies and Atomic Energy Commission (CEA), France Génomique (ANR-10-INBS-09-08, DYNAMO project), the Centre de coopération Internationale en Recherche Agronomique pour le Développement (CIRAD), and the Agropolis Fondation (ID 1504-006) 'GenomeHarvest' project through the French Investissements d'Avenir program (Labex Agro: ANR-10-LABX-0001-01). We thank CIRAD-INRAE Biological Resource Centre for Tropical Plants (CRB-PT) in the West Indies (Guadeloupe, France) for providing the plant materials. We are grateful to Jean-Marie Delos, Franck Marius and Jean-Claude Efile for their contributions to progeny development. This work was technically supported by the CIRAD—UMR AGAP HPC Data Center of the South Green Bioinformatics platform.

## Author contributions

CG, FS and JM developed and provided plant materials. KL and CC managed library preparation and genome sequencing. BI, CB and JMA carried out the assembly of the genomes. BN performed the gene prediction. GM performed most global genome comparison, in silico chromosome ancestry painting and phylogenetic analyses. FCB performed TE and repeated tandem analysis. NY performed local phylogenetic analysis. CH performed cytogenetic analysis. CG and FS developed progenies. JMA and CB submitted the data. FA and HAV provided sequence data access. MR, GD integrated data in the Banana Genome Hub. GM, AD, NY, FCB, JMA wrote the draft manuscript that was then improved by MR, JS. AD, JMA, GM NY, PW conceived the project. AD and JMA supervised the study.

## Competing interests

The authors declare no competing interests.
