## [Peer Review file · Nature Communications]

Unravelling genomic drivers of speciation in *Musa* through genome assemblies of wild banana ancestors

Corresponding Author: Dr Guillaume Martin

Version 0:

Reviewer comments:

Reviewer #1

(Remarks to the Author)

The banana genome research, particularly concerning cash crops, has undergone rapid development. Not only has there been significant progress in comparative genome studies, but also in-depth investigations into the key genes responsible for trait formation. The work obtained diploid banana genomes are limited in number, and there have been no substantial recent findings in comparative genomics studies of bananas. Although some diploid genomes and two haplotypes have been identified, along with insights into the origin of cultivated bananas, few new results have been found compared to the previously published cultivated banana genome.

Reviewer #2

(Remarks to the Author)

The authors reported four wild genetic *Musa* species and one hybrid diploid cultivar banana genome with two unknown ancestral contributors. These assemblies are complementary with the published banana genomes, providing access to reference genomes for the main contributors to major banana cultivars. They also deduced the two unknown contributors. Global comparative genome and phylogenetic analyses suggested an ongoing speciation process within the *Musa* lineage through large chromosome rearrangements. This study has certain significance, but the following issues need to be addressed.

Major

1. The weakness of this study is the lack of biological stories. Currently, genome articles need to be analyzed in conjunction with biological issues, and the author needs to include appropriate biological stories in this manuscript.
2. Most of the genomes were sequenced with Oxford Nanopore Technologies (ONT), which is not as accurate as PacBio sequencing. The authors should further validate the accuracy of genome assemblies with multiple methods and data, including the Illumina sequencing reads and RNA-seq data.
3. The recently published *Musa textilis* (Abaca) genome has an assembly size of 613,062,180 bp, with 35,077 annotated genes and 58.41% TE content. And the genome size of *M. textilis* in this study is 545,596,533 bp, which is significantly different from the published ones. What are the sequences of these difference? In addition, the number of annotated genes is 33,662, and the annotated TE content is 62.18%. Compared to published genomes, the number of annotated genes is less, while the TE content is more. Please make a detailed comparison and explain the reasons for these differences.
4. In the reference (Li et al. Nat Genet. 2024, 56(1):136-142), the genome size of *M. zebrina* is 548,070,311 bp and the number of genes is 31,049. The genome size of the published *M. banksii* was 464,148,587 bp, with a total of 32,682 genes. Please compare and analyze the above genomes with the assembled genomes in this analysis. In addition, the genome size of *M. burmannica* is 526,296,796 bp with a number of genes of 45,044, while the upgraded *M. burmannica* genome in this study is 505,072,535 bp, which has a shorter genome length and fewer genes. Please compare the specific difference in details.
5. How to determine whether SIRE/Maximus is located in the near centromere region or the centromere region in this analysis? The distribution range of SIRE/Maximus elements is very wide in Figure 3a and 3b, and some even occupy half of the chromosomes, while the centromere clearly does not have such a large proportion. Therefore, the statement mentioned in this manuscript that SIRE/Maximus is located in the near centromere region or the centromere region needs further clarified.

Minor

6. In Supplementary Methods Fig.2: Normalized curve ratio for each read set of Pisang Madu, the horizontal and vertical coordinates have not provided.
7. The legend in Figures 3a/3b shows the use of gray to represent SIRE/Maximus elements, but gray is not found in Figures 3a/3b, and Supplementary Fig. 1 is the same situation.
8. The halabanensis (P. Madu H2) in Figure 4 should be modified to halabanensis (P. Madu H1).
9. In line 234, banksii (26% of the assembly), zebrina (24%), malaccensis (20%), the unknown contributor (11%), schizocarpa (1%), and possibly halabanensis (1%) 235 (Figure 6), all these percentages add up to 83%, which is not 100%. Which ancestor diploid banana did the remaining portion come from?
10. The first large translocation variation in Figure 6d has a color representation consistent with Figure 2. From Figure 2, it can be seen that red represents *M. zebrina*, green represents *M. banksii*, and there is a portion of green in chr3T8 that is the source of *M. banksii*, indicating that the Cavendish region has been identified as originating from *M. banksii*. However, the author suggests that chr3T8 belongs to the translocation between *M. zebrina* and Cavendish, why the existence of *M. banksii* has been ignored?

Reviewer #3

(Remarks to the Author)

Martin et al. generated (or improved) chromosome-scale assemblies for seven *Musa* genomes, comprising five diploid *Musa* species or subspecies and the genomes of a diploid hybrid cultivar, related to the cultivated triploid Cavendish cultivar. Furthermore, they characterized the distribution of the main tandem and dispersed repeats along the 11 (or 10) chromosomes of the tested *Musa* accessions. The obtained data together with previous ones were used to establish phylogenetic trees and synteny relationships. These results elucidated the evolution of *Musa* genomes and enabled the authors to identify the main (including two previously unknown) contributors to the triploid Cavendish genome, as well as multiple rearrangements between and within the related genomes of the tested accessions.

The manuscript provides an essential contribution to the ancestry and evolution of an important crop and obliterates many white spots in the complex ancestry of domesticated banana. Because of its high density, the paper is not easy to grasp for readers without detailed knowledge of banana diversity. It could be made more reader-friendly, in particular the illustrations. Including more taxonomy (and phylogeny?) in the first line of table 1 would be very helpful for non-experts (not just *zebrina*, *banksii*...). The taxonomic terms in Figs. 2 and 4 should have a larger font size to become readable. The term 'chromosome painting' is already occupied for a distinct cytogenetic technique. To demonstrate the ancestral composition of chromosomes, another term, e.g., 'chromosome ancestry', could be used instead throughout the text. The figure legends could be more explanatory.

Why are three accessions enframed in Fig. 2 c, d? What means (GTR+ Γ model) on line 733?

What do the arrows in Fig. 3d indicate? The inserts in Fig. 2c-e shows a chromosome belonging to the remaining metaphase complement? According to 2b, all but one pair of *M. schizocarpa* chromosomes should display 5S rDNA signals? Do the authors have indication about the transcriptional activity (e.g., AgNO₃-staining) of (peri)centromeric rDNA loci?

In the tree of Fig. 4 shouldn't it read: halabanensis (P. Madu H1) as in Table 1? Also here, and in other figures, the taxonomic status of the accession should be given.

On line 33 'taxonomic origin' should be substituted by 'phylogenetic position'.

On lines 110 and 352 it should read 'diploid hybrid' instead of 'hybrid diploid'.

On line 187 iii) is missing.

On line 216 the authors could mention that chromosomes 7,8,9 of *M. textilis* (n=10) correspond the chromosomes 7,8,9,11 of *M. balbisiana* and *M. halabanensis* (both n=11), explaining the different chromosome numbers on the basis of synteny (Fig. 4).

In the Discussion section, the authors should refer to figures where ever possible. For instance, I find a 45S rDNA cluster on chromosome 10 only for *M. a. ssp. zebrina*, but not for *M. schizocarpa* (lines 308/9; Fig. 3b).

On line 304 it should read: 'gametes ... are aneuploid,' (not aneuploids).

On line 365 it should read: burmannica.

Ingo Schubert

Version 1:

Reviewer comments:

Reviewer #1

(Remarks to the Author)

fig2's Madu H2 looks like most of the ancestors came from *banksii*, as well as an unknown. But in fig3, Madu H2 is close to *zebrina*, which doesn't make sense. Unless this unknown itself has most of the ancestors from *Zebrina*.

It is best to compare the ancestral components of Madu H2 in fig2 with the data with the assembled genome, so it is recommended to remove the "unknown" component and use the assembled genome to compare. "unknown" itself is so close to *zebrina* that it is difficult to distinguish whether the Madu H2 comes from "unknown" or from *zebrina*. This means that it is better to use the genomic data in fig3 to distinguish the origin of the Madu H2 ancestry, rather than using an unknown sample without a genome.

If the Madu H2 ancestor is half zebrina and half is from banksii, fig3 tree shows that this seems to be the result. Using the "unknown" to distinguish the ancestors of the Cavendish banana is to bring more noise. It is better to distinguish the ancestors of Cavendish banana directly by banksii and zenrina. Most of the better explanations the ancestors of Cavendish banana from Zebrina, banksii and DH. So using the "unknown" to discuss the composition of the Cavendish banana only creates more misunderstandings and does not give more or better explanations than the work of Li et al. So I suggest cutting out the section comparing the ancestral origins of the Cavendish banana.

Reviewer #2

(Remarks to the Author)

Although the authors have addressed most of my concerns, and this manuscript make some contributions to the field of evolutionary biology. I still think the new results provided in this paper are not enough to be published in Nature Communications.

Reviewer #3

(Remarks to the Author)

The revised version improved the manuscript much, albeit it was difficult to trace the changes announced in the letter of response, and the answer regarding the transcriptional activity of the (peri)centromeric rDNA loci is pretty vague.

Still there is some sloppiness to be corrected in a final version:

Line 95: delete the second 'recent'

Line 123: Is the highest BUSCO value 98.7 or 98.9 (Table 1)?

Lines 124/5: the gene numbers are not identical with those in Tab.1.

Line 243: it should read (Fig. 5d,e) (delete c.)

Line 372: insert (Fig. 5a,b).

Line 374: omit "suggesting that...", because the few cases and comparison with other chromosomes involved do not allow to conclude that acrocentrics are 'less stable'.

Line 473: Not Musa schizocarpa but its genome assembly was improved.

Lines: 698 and 706: substitute 2023 by 2024

Line 888: substitute (a) by (c)

Ingo Schubert

Version 2:

Reviewer comments:

Reviewer #1

(Remarks to the Author)

1. The "unknown" genome is very important, and many of the results of this study are unreliable without using the "unknown" genome to study its proximity to Zebrina or banksii. Therefore, it is necessary to assemble the "unknown" genome.

2. "It is best to compare the ancestral components of Madu H2 in fig2 with the data with the assembled genome, so it is recommended to remove the "unknown" component and use the assembled genome to compare." Similarity comparison with "unknown" second-generation sequence alignment will lead to a lot of errors or false positives. It is necessary to use the genome for comparison. So the above work is to be done.

Reviewer #3

(Remarks to the Author)

I am now satisfied by the response of the authors.

Ingo Schubert

Version 3:

Reviewer comments:

Reviewer #1

(Remarks to the Author)

1. According to fig3, P. Madu H2 has five chromosomes like M.a. banksii and no chromosome like M.a. zebrina, but P. Madu H2 and M.a. Zebrina, on the contrary, is the closest to each other. Two species that are close to each other do not have chromosomes that are similar, but are similar to those who are outside with chromosomes. So fig3 is a chromosomal similarity comparison is unreasonable and unacceptable.

So I understand that the unknown sample in "P. Madu H2" should be "M.a. zebrina", or the closest to "M.a. zebrina".

Therefore, it is necessary to distinguish the relationship between the unknown sample and "M.a. zebrina", which is the focus of this study. Many of the conclusions of this study are unreliable if they cannot be distinguished.

As I suggested earlier, the genome of the unknown sample should be obtained, but it is not available, so I recommend removing the unknown.

2. Li et al. only analyzed the possible ancestors of the Cavendish genome, and indeed the ancestors may have come from three parents, similar to fig6 results, and did not analyze content such as chromosome exchange, as well as the focus on gene mining.

So it is not "However, none of these rearrangements were found in the Cavendish assembly of Li et al".

Li et al. are important works to study the origin and gene mining of Cavendish.

3. fig5 shows that there are many repeats in the genome, which corresponds to Sup fig8, indicating that most of the regions that are not aligned or not linear are repeats or centromeres.

Dear reviewers,

Thank you for your comments that have helped us to improve our manuscript entitled “Unravelling genomic drivers of speciation in *Musa* through genome assemblies of wild banana ancestors”. Please find our response hereafter (in blue).

Sincerely,
Dr. Guillaume Martin & Dr Angélique D’Hont

Responses to comments:

Reviewer #1 (Remarks to the Author):

The banana genome research, particularly concerning cash crops, has undergone rapid development. Not only has there been significant progress in comparative genome studies, but also in-depth investigations into the key genes responsible for trait formation. The work obtained diploid banana genomes are limited in number, and there have been no substantial recent findings in comparative genomics studies of bananas. Although some diploid genomes and two haplotypes have been identified, along with insights into the origin of cultivated bananas, few new results have been found compared to the previously published cultivated banana genome.

As highlighted above, our manuscript makes substantial contributions to the field of evolutionary biology. Specifically, our study focuses on the speciation process through two distinct mechanisms of chromosome evolution: large rearrangements and centromere differentiation. We have revised our manuscript to emphasize this novel insight. Furthermore, we have contributed a complete assembly of a previously unknown ancestor and a partial assembly of a still unknown ancestor and used this information to pinpoint the phylogenetic positions of these two previously unknown ancestors of cultivated bananas, corresponding to germplasms that are not available in any ex situ collections worldwide. These findings and associated genome information are a significant advancement for the field of conservation genomics.

Reviewer #2 (Remarks to the Author):

The authors reported four wild genetic *Musa* species and one hybrid diploid cultivar banana genome with two unknown ancestral contributors. These assemblies are complementary with the published banana genomes, providing access to reference genomes for the main

contributors to major banana cultivars. They also deduced the two unknown contributors. Global comparative genome and phylogenetic analyses suggested an ongoing speciation process within the *Musa* lineage through large chromosome rearrangements. This study has certain significance, but the following issues need to be addressed.

Major

1. The weakness of this study is the lack of biological stories. Currently, genome articles need to be analyzed in conjunction with biological issues, and the author needs to include appropriate biological stories in this manuscript.

As highlighted above, while our manuscript does not follow the traditional structure of a genome assembly paper, which typically includes an analysis of a targeted metabolic process, it makes significant contributions to the field of evolutionary biology. Specifically, our study focuses on the speciation process through two distinct mechanisms of chromosome evolution: large rearrangements and centromere differentiation. We revised our manuscript in order to better reflect these findings and think that we are now presenting a more compelling story on the evolution of bananas. By doing so, we extensively revised the discussion sections and added subtitles to convey key messages. This approach better guides readers to the biological insights and creates a more cohesive narrative flow.

2. Most of the genomes were sequenced with Oxford Nanopore Technologies (ONT), which is not as accurate as PacBio sequencing. The authors should further validate the accuracy of genome assemblies with multiple methods and data, including the Illumina sequencing reads and RNA-seq data.

PacBio and ONT sequencing are now the standard methods for generating high-quality assemblies, often requiring integration with long-range data such as Hi-C data or optical maps. In this study, we employed these standard methods, and our assembly meets the quality criteria set by the Earth BioGenome Project (EBP). The lower accuracy of nanopore sequencing was offset by higher coverage and the use of Illumina short-reads. As suggested by the reviewer, we utilized Illumina reads from DNA and RNA libraries to estimate the quality of our assemblies (Supplementary Methods Table S3). We found no significant differences between our ONT and PACBIO assemblies, except for the Merqury QV score (Supplementary Result 1). It's important to note that for ONT assemblies, this score was obtained using an independent dataset (e.g., Illumina short-reads), whereas for PacBio assemblies, the same PacBio data was used, leading to an overestimation of these scores. The recent ERGA preprint highlights this critical point (<https://doi.org/10.1101/2023.09.25.559365>) but shows that both technologies allow generating assemblies that meet EBP recommendations.

We also compared our assemblies with available ones, and overall, our assemblies are of higher quality (due to long-reads) or equivalent quality (Supplementary Result 1).

3. The recently published *Musa textilis* (Abaca) genome has an assembly size of 613,062,180 bp, with 35,077 annotated genes and 58.41% TE content. And the genome size of *M. textilis* in this study is 545,596,533 bp, which is significantly different from the published ones. What are the sequences of these differences? In addition, the number of annotated genes is 33,662, and the annotated TE content is 62.18%. Compared to published genomes, the number of annotated genes is less, while the TE content is more. Please make a detailed comparison and explain the reasons for these differences.

The *Musa textilis* 'ABACA' (Zhou et al., 2024) genome is composed of 613,062,180 bp with 606,501,102 bp of chromosome assembly and 6,561,078 pb of unanchored scaffolds and was based on another accession of *Musa textilis* from the Bioversity International Transit Center in Leuven (Belgium). ABACA assembly is indeed slightly larger (613 Mb vs 545 Mb, ABACA being 13% greater in size).

Assemblies comparison was performed using a dot plot approach revealing a global synteny between the two assemblies. Most chromosomes of the ABACA assembly are longer than the corresponding ones in our study. Almost all ABACA chromosomes contained kilobase long duplications that were not present in the *Musa textilis* assembly from our study. Based on comparative data (from Supplementary Results 1), our *Musa textilis* assembly is slightly less complete (BUSCO 96.9% vs 95.8%) but also less duplicated (3.9% vs 9.2%), Mercury score and mercury completeness score are better for our *Musa textilis* assembly. KAT plots analysis reveal that both genomes are diploid and heterozygous. However, the ABACA genome assembly still includes allelic duplications and surprisingly a high proportion of missing kmers.

These results have been added in the supplementary result 1 section of the manuscript.

A noticeable exception in the synteny between these genomes was the end of chromosome 01 in which a 6 Mb fragment was absent in our assembly. We thank the reviewing process for pointing out this error. The fragment has been reintegrated in a new version of assembly that will be available in the banana genome Hub repository.

4. In the reference (Li et al. Nat Genet. 2024, 56(1):136-142), the genome size of *M. zebria* is 548,070,311 bp and the number of genes is 31,049. The genome size of the published *M. banksii* was 464,148,587 bp, with a total of 32,682 genes. Please compare and analyze the above genomes with the assembled genomes in this analysis. In addition, the genome size of *M. burmannica* is 526,296,796 bp with a number of genes of 45,044, while the upgraded *M. burmannica* genome in this study is 505,072,535 bp, which has a shorter genome length and fewer genes. Please compare the specific difference in details.

The *M. a. zebrina* assembly from Li et al 2024 is similar in size to our *zebrina* assembly, with the former being 548 Mb and the latter 539 Mb. Assemblies comparison using a dot plot approach revealed a global synteny between the two assemblies except for i) the translocation between chromosomes 03 and 08 not present in Li et al assembly and ii) a 7Mb fragment corresponding to chromosome 11 centromere is lacking in the Li et al assembly. Li et al 2024 guided their assembly with the published DH-Pahang assembly and thus kept the chromosomal structure of DH-pahang and so this assembly did not display the 3/8 translocation although all the *M. a. zebrina* accessions analyzed so far displayed this translocation. In the absence of passport data for the *M. a. zebrina* clone used by these authors, it is thus not possible to verify if this accession contains this translocation, however it is expected.

The two other mentioned assemblies from *M. a. burmannica* and *M. a. banksii* were preliminary assemblies that we produced and published in Rouard et al 2018. These assemblies remained as scaffold assemblies and not chromosome scale assemblies and the gene annotation was fragmented. It is thus difficult to compare their size and gene numbers

but the trends of genome sizes between subspecies are conserved and our assemblies are improved and of higher quality.

5. How to determine whether SIRE/Maximus is located in the near centromere region or the centromere region in this analysis? The distribution range of SIRE/Maximus elements is very wide in Figure 3a and 3b, and some even occupy half of the chromosomes, while the centromere clearly does not have such a large proportion. Therefore, the statement mentioned in this manuscript that SIRE/Maximus is located in the near centromere region or the centromere region needs further clarified.

Thank you, this point has now been corrected in the manuscript, SIRE/Maximus is clearly abundant in pericentromeric regions while centromeric regions are tagged by the presence of the Nanica LINE element. CRM elements that have been shown to cluster in centromeric regions in other plants (doi: 10.1105/tpc.006106) have been added to the figure for a better visualization of centromeric regions.

Minor

6. In Supplementary Methods Fig.2: Normalized curve ratio for each read set of Pisang Madu, the horizontal and vertical coordinates have not provided.

The vertical and horizontal coordinates were added.

7. The legend in Figures 3a/3b shows the use of gray to represent SIRE/Maximus elements, but gray is not found in Figures 3a/3b, and Supplementary Fig. 1 is the same situation.

The grey color was homogenized between figures and legends.

8. The halabanensis (P. Madu H2) in Figure 4 should be modified to halabanensis (P. Madu H1).

This was corrected.

9. In line 234, banksii (26% of the assembly), zebrina (24%), malaccensis (20%), the unknown contributor (11%), schizocarpa (1%), and possibly halabanensis (1%) 235 (Figure 6), all these percentages add up to 83%, which is not 100%. Which ancestor diploid banana did the remaining portion come from?

The remaining proportion could not be attributed to an ancestor. It probably corresponds, in part, to the unknown contributor but as a complete genome of this unknown contributor is not yet available, determining the origin of all segments from the unknown ancestor is challenging. Another part may represent repeat-rich pericentromeric regions which also complicates origin prediction.

10. The first large translocation variation in Figure 6d has a color representation consistent with Figure 2. From Figure 2, it can be seen that red represents *M. zebrina*, green represents *M. banksii*, and there is a portion of green in chr3T8 that is the source of *M. banksii*, indicating that the Cavendish region has been identified as originating from *M. banksii*. However, the author suggests that chr3T8 belongs to the translocation between *M. zebrina* and Cavendish, why the existence of *M. banksii* has been ignored?

Chromosome chr08-h2 in Cavendish results from a recombination between a *M. a. ssp zebrina* chromosome (chr3T8) and a *M. a. ssp banksii*. Because in this chromosome chr08-h2, the region corresponding to the 3T8 translocation breakpoint corresponds to a region originating from *M. a. ssp zebrina*, this chromosome in Cavendish is expected to have the 3T8 translocated structure.

Reviewer #3 (Remarks to the Author):

Martin et al. generated (or improved) chromosome-scale assemblies for seven *Musa* genomes, comprising five diploid *Musa* species or subspecies and the genomes of a diploid hybrid cultivar, related to the cultivated triploid Cavendish cultivar. Furthermore, they characterized the distribution of the main tandem and dispersed repeats along the 11 (or 10) chromosomes of the tested *Musa* accessions. The obtained data together with previous ones were used to establish phylogenetic trees and synteny relationships. These results elucidated the evolution of *Musa* genomes and enabled the authors to identify the main (including two previously unknown) contributors to the triploid Cavendish genome, as well as multiple rearrangements between and within the related genomes of the tested accessions.

The manuscript provides an essential contribution to the ancestry and evolution of an important crop and obliterates many white spots in the complex ancestry of domesticated banana. Because of its high density, the paper is not easy to grasp for readers without detailed knowledge of banana diversity. It could be made more reader-friendly, in particular the illustrations.

In order to make the manuscript more reader-friendly, we simplified the text when possible, added some general context, added more comprehensive subtitles in the result section, revised the discussion section, modified figures in particular by making classification in species and subspecies more evident and improved the figure legends.

1) Including more taxonomy (and phylogeny?) in the first line of table 1 would be very helpful for non-experts (not just *zebrina*, *banksii*...).

Thank you for the suggestion. A taxonomy line has been added at the top of Table 1 to indicate the species represented by each assembly.

2) The taxonomic terms in Figs. 2 and 4 should have a larger font size to become readable.

Thank you for the suggestion. We modified the figures font size.

3) The term 'chromosome painting' is already occupied for a distinct cytogenetic technique. To demonstrate the ancestral composition of chromosomes, another term, e.g., 'chromosome ancestry', could be used instead throughout the text.

We changed the expression of "chromosome painting" to "*in-silico* chromosome ancestry painting".

4) The figure legends could be more explanatory.

Why are three accessions enframed in Fig. 2 c, d?

The following sentence was added in the figure 2 legend. "Sequences that are grouped in polytomy together with *Musa a. ssp halabanensis* are enframed in **c** and **d**."

5) What means (GTR+ Γ model) on line 733?

This corresponds to the General Time Reversible (GTR) model across lineages along with a gamma (+ Γ) distributed rate across sites. We modified the Material and method section to make this acronym used in the legend more understandable.

6) What do the arrows in Fig. 3d indicate?

Arrows indicate clusters of 45S rDNA of smaller size and thus detected with lower signal compared to the major 45S rDNA. The Figure legend was modified to explain these arrows.

7) The inserts in Fig. 2c-e shows a chromosome belonging to the remaining metaphase complement?

We guess it concerns "Fig. 3c-e". The inserts represent a chromosome that was offset from the presented field of view. This was added to the figure legend.

8) According to 2b, all but one pair of *M. schizocarpa* chromosomes should display 5S rDNA signals?

We guess it concerns "3b". Most but not all 5S sites detected *in silico* on our assembly were also detected in centromeric regions of *Musa schizocarpa* using *FISH*. Our assembly contained 12 sites with a substantial amount of 5S rDNA sequences whereas 10 sites with various intensity are visible on the picture. Because some sites contain less repeats they may be more difficult to detect. Our main purpose was to verify that this finding of rDNA sequence in the centromeric region of many chromosomes, that is unusual, was not due to assembly problems.

9) Do the authors have indication about the transcriptional activity (e.g., AgNO₃-staining) of (peri)centromeric rDNA loci?

We made some experiments that did not seem to reveal major transcriptional activity in centromeric regions of *Musa schizocarpa* but the results were not clear enough to definitively conclude.

10) In the tree of Fig. 4 shouldn't it read: halabanensis (P. Madu H1) as in Table 1? Also here, and in other figures, the taxonomic status of the accession should be given.

Thank you for raising this point. P. Madu H2 was corrected to P. Madu H1 on Fig. 4. We modified figures in order to make species and subspecies of analyzed genomes classification more reader-friendly.

11) On line 33 'taxonomic origin' should be substituted by 'phylogenetic position'.

This was changed.

12) On lines 110 and 352 it should read 'diploid hybrid' instead of 'hybrid diploid'.

This was corrected.

13) On line 187 iii) is missing.

This was corrected.

14) On line 216 the authors could mention that chromosomes 7,8,9 of *M. textilis* (n=10) correspond the chromosomes 7,8,9,11 of *M. balbisiana* and *M. halabanensis* (both n=11), explaining the different chromosome numbers on the basis of synteny (Fig. 4).

We modified the sentence in this paragraph.

15) In the Discussion section, the authors should refer to figures where ever possible. For instance, I find a 45S rDNA cluster on chromosome 10 only for *M. a. ssp. zebrina*, but not for *M. schizocarpa* (lines 308/9; Fig. 3b).

We added a pink asterisk in Fig. 3b to indicate that the region contains a stretch of N impacting the full representation of the rDNA cluster.

16) On line 304 it should read: 'gametes ... are aneuploid,' (not aneuploids).

This was corrected

17) On line 365 it should read: burmannica.

This was corrected

Ingo Schubert

REVIEWER COMMENTS

Reviewer #1 (Remarks to the Author):

fig2's Madu H2 looks like most of the ancestors came from banksii, as well as an unknown. But in fig3, Madu H2 is close to zebrina, which doesn't make sense. Unless this unknown itself has most of the ancestors from Zebrina.

Figure 2 shows that P. madu H2 haplotype is a mosaic of ancestral contribution predominantly composed of banksii and unknown (Figure 2b) as rightly noted by the reviewer. The GBBS gene, which clusters with ssp *banksii* gene in the phylogeny (Figure 2d), is located in a region of banksii origin in the P. Madu H2 assembly, which is consistent (this has been added in the figure). The ADH gene, which cluster with ssp *zebrina* in the phylogeny (Figure 2c), is located in a region of unknown origin in the P. Madu H2 (this has been added in the figure), which is consistent with ssp *zebrina* being a close relative of the unknown origin.

The phylogeny in Figure 3 was thus made using only the genes corresponding to the unknown ancestor in P. madu H2 (and their orthologs in the other assemblies), that is why P. Madu H2 in this phylogeny is close to zebrina and not to banksii.

It is best to compare the ancestral components of Madu H2 in fig2 with the data with the assembled genome, so it is recommended to remove the "unknown" component and use the assembled genome to compare. "unknown" itself is so close to zebrina that it is difficult to distinguish whether the Madu H2 comes from "unknown" or from zebrina. This means that it is better to use the genomic data in fig3 to distinguish the origin of the Madu H2 ancestry, rather than using an unknown sample without a genome.

According to the phylogeny on figure 3, the divergence between the unknown ancestor and *M.a. ssp. zebrina* is in a comparable range to the divergence between *M. a. halabanensis* from *M. schizocarpa* and *M. a. malaccensis* from *M. a. burmannica* so we think it is important to distinguish the unknown ancestor from *M. a. zebrina*.

The existence of this unknown ancestor has been published as well as the ability using in-silico ancestry chromosome painting to differentiate the ancestral contributions from *M. a. ssp zebrina* and the unknown ancestor:

- *Martin, G. et al. Interspecific introgression patterns reveal the origins of worldwide cultivated bananas in New Guinea. Plant J. 113, 802–818 (2023).*
- *Sardos, J. et al. Hybridization, missing wild ancestors and the domestication of cultivated diploid bananas. Front. Plant Sci. 13, (2022).*

If the Madu H2 ancestor is half zebrina and half is from banksii, fig3 tree shows that this seems to be the result. Using the "unknown" to distinguish the ancestors of the Cavendish banana is to bring more noise. It is better to distinguish the ancestors of Cavendish banana directly by banksii and zenrina. Most of the better explanations the ancestors of Cavendish banana from Zebrina,banksii and DH. So using the "unknown" to discuss the composition of the Cavendish banana only creates more misunderstandings and does not give more or better explanations than the work of Li et al. So I suggest cutting out the section comparing the ancestral origins of the Cavendish banana.

Since we have shown that Cavendish does not only have contributions from *banksii*, *zebrina* and *malaccensis*, we think it is important to keep the section about the ancestral origins of Cavendish banana. For example, if we are able to show that important characters were contributed by the unknown ancestor, efforts can be made to identify the corresponding wild material so that breeders can use these materials as sources for these traits. If the ancestral origin of the traits is not correctly identified, breeding efforts may be misdirected.

Reviewer #2 (Remarks to the Author):

Although the authors have addressed most of my concerns, and this manuscript make some contributions to the field of evolutionary biology. I still think the new results provided in this paper are not enough to be published in Nature Communications.

We are grateful for the time and effort invested in reviewing our manuscript. The feedback provided has been truly helpful in improving our work.

Reviewer #3 (Remarks to the Author):

The revised version improved the manuscript much, albeit it was difficult to trace the changes announced in the letter of response, and the answer regarding the transcriptional activity of the (peri)centromeric rDNA loci is pretty vague.

Sorry for not having been clear enough, we do not have access to fresh roots of *M. schizocarpa* to perform cytogenetic experiments at the moment, so we made some AgNO₃-staining experiments (we had not previously done this type of staining) with the remaining slides we had and the results were not very clear: no clear signals were obtained even for the main rDNA clusters. We should renew this experiment in the future with fresh material when available.

Still there is some sloppiness to be corrected in a final version:

Line 95: delete the second 'recent'

This has been corrected.

Line 123: Is the highest BUSCO value 98.7 or 98.9 (Table 1)?

Value has been corrected to 98.9 as in Table 1 and Supplementary METHODS Table 3.

Lines 124/5: the gene numbers are not identical with those in Tab.1.

Correct values are those of the table. Value has been corrected in the text.

Line 243: it should read (Fig. 5d,e) (delete c,)

Figure 5c shows the location of Nanica-type transposable elements, which are landmarks of Musa centromeres, while Figures 5d and 5e show the location of the 45S and 5S sequences respectively. Figure 5c is therefore necessary to show that the 45S and 5S sequences colocalize with the Nanica sequences and are therefore also in the centromeric region. We modified the sentence to refer to the Fig5c.

Line 372: insert (Fig. 5a,b).

This has been corrected.

Line 374: omit "suggesting that...", because the few cases and comparison with other chromosomes involved do not allow to conclude that acrocentrics are 'less stable'.

We have removed the paragraph and added line 234 that three chromosomes are acrocentric.

Line 473: Not *Musa schizocarpa* but its genome assembly was improved.

This has been corrected.

Lines: 698 and 706: substitute 2023 by 2024

This has been corrected.

Line 888: substitute (a) by (c)

This has been corrected.

Ingo Schubert

REVIEWER COMMENTS

Reviewer #1 (Remarks to the Author):

1. The "unknown" genome is very important, and many of the results of this study are unreliable without using the "unknown" genome to study its proximity to *Zebrina* or *banksii*. Therefore, it is necessary to assemble the "unknown" genome.

To our knowledge, the "unknown" genome has no wild pure representatives in international genebanks. Furthermore, none of the hybrids tested so far has a complete haplotype of this "unknown" ancestor in its genome. Therefore, it is not possible at this stage to assemble the complete genome of the unknown ancestor.

Pisang Madu is one of the individuals with the highest proportion of this "unknown" ancestor in its genome (<https://onlinelibrary.wiley.com/doi/full/10.1111/tpj.16086>). Thus, to gain access to at least part of this genome, we assembled the two haplotypes of Pisang Madu, then we identified the chromosome segments corresponding to this "unknown" ancestor in this assembly and carried out phylogeny studies (Fig.3) using only the genes present in the chromosome segments contributed by the "unknown" ancestor. Thus, although it is not possible at this stage to assemble the whole unknown genome, we have assembled part of this genome (the part present in P. Madu haplotype2). Because we used only the genes present in the chromosome segments contributed by the "unknown" ancestor in haplotype 2 of Pisang Madu, we could place this genome in our phylogenetic analysis and showed that it is distinct but close to the *zebrina* subspecies of *Musa acuminata*.

2. "It is best to compare the ancestral components of Madu H2 in fig2 with the data with the assembled genome, so it is recommended to remove the "unknown" component and use the assembled genome to compare." Similarity comparison with "unknown" second-generation sequence alignment will lead to a lot of errors or false positives. It is necessary to use the genome for comparison. So the above work is to be done.

If we understand correctly, reviewer 1 proposes to perform assembly ancestry painting using an approach similar to the one described in Li et al., 2024. Doing so, we would use the genome assembly of subspecies *banksii*, *zebrina*, *malaccensis*, *burmannica*, *halabanensis*, and other *Musa* species assemblies to paint the Pisang Madu haplotype. Doing so, if an ancestry present in the Pisang Madu haplotype is not represented in the assemblies datasets used to determine local ancestry, the origin will be attributed to its closest relative present in the dataset. Thus, using this approach, it will not be possible to identify the chromosome segments contributed by the unknown genome, instead these segments would likely be wrongly attributed to its closest relative which is *M. a. ssp zebrina*. The presence of unknown ancestors, in Pisang Madu but also in other banana cultivars, including Cavendish, was demonstrated in peer reviewed work (Martin et al., 2020, Sardos et al., 2022, Martin et al., 2023); as well as the fact that this unknown ancestor in Pisang Madu is distinct from *M. a. ssp zebrina* (Martin et al., 2023).

Reviewer 1 suggests that second-generation sequence alignment will lead to a lot of errors or false positives. Tags used to perform chromosome ancestry painting in our approaches are illumina reads of 150 nucleotides that have a very low error rate and only exact matches (mismatches and indels not authorised) were used to attribute ancestry. The use of assembled genomes that are a consensus of two haplotypes could also induce bias in the analysis.

Reviewer #3 (Remarks to the Author):

I am now satisfied by the response of the authors.

Ingo Schubert

We would like to thank Professor Schubert for the time and effort he put into reviewing our manuscript. The feedback provided was really helpful in improving our manuscript.

REVIEWER COMMENTS

Reviewer #1 (Remarks to the Author):

1. According to fig3, P. Madu H2 has five chromosomes like *M.a. banksii* and no chromosome like *M.a. zebrina*, but P. Madu H2 and *M.a. Zebrina*, on the contrary, is the closest to each other. Two species that are close to each other do not have chromosomes that are similar, but are similar to those who are outside with chromosomes. So fig3 is a chromosomal similarity comparison is unreasonable and unacceptable.

So I understand that the unknown sample in "P. Madu H2" should be "*M.a. zebrina*", or the closest to "*M.a. zebrina*".

Therefore, it is necessary to distinguish the relationship between the unknown sample and "*M.a. zebrina*", which is the focus of this study. Many of the conclusions of this study are unreliable if they cannot be distinguished.

We think there is misunderstanding. In our three previous responses, we have done our best to address the same concerns of reviewer 1 about the existence of the 'unknown' ancestor, and we have also modified the figures to enhance clarity.

The existence of the unknown ancestor and its distinction from *M. a. ssp zebrina* has been validated in three peer-reviewed papers (Martin et al. 2020, doi.org/10.1111/tpj.14683; Sardos et al. 2022, doi.org/10.3389/fpls.2022.969220; Martin et al. 2023, doi.org/10.1111/tpj.16086).

As I suggested earlier, the genome of the unknown sample should be obtained, but it is not available, so I recommend removing the unknown.

The term "unknown" seems appropriate to us, and the unknown genome could not be removed, it is part of important cultivars such as Cavendish.

2. Li et al. only analysed the possible ancestors of the Cavendish genome, and indeed the ancestors may have come from three parents, similar to fig6 results, and did not analyse content such as chromosome exchange, as well as the focus on gene mining.

So it is not "However, none of these rearrangements were found in the Cavendish assembly of Li et al".

Li et al. are important works to study the origin and gene mining of Cavendish.

We agree that the assembly produced by Li et al. (2023) is important to access the genes of Cavendish but regarding the ancestral origin of Cavendish and the related chromosome structures involved, we think that this assembly could be improved in the future and we provide keys to do so.

Cavendish has 3 large described reciprocal translocations (Martin et al. 2020 - doi.org/10.1111/tpj.15031) that lack in Li et al. (2023) assembly and are partially present in assembly of Huang et al. (2023).

To improve these assemblies in the future, we propose a strategy that exploits ancestry chromosome painting to identify contigs/haplotype origin. This origin can then be used to select the correct assembly to reference guide each haplotype and thus fully reproduce the chromosome structure of Cavendish.

3. fig5 shows that there are many repeats in the genome, which corresponds to Sup fig8, indicating that most of the regions that are not aligned or not linear are repeats or centromeres.

Indeed, banana genomes contain approximately 50% repeat sequences, most of which are located in centromeric regions, which could explain part of the differences observed but not all.